# Dissipative quantum error correction and application to quantum sensing with trapped ions

F. Reiter [1,2,3], A.S. Sørensen [4], P. Zoller[1,2] & C.A. Muschik[1,2]

Quantum-enhanced measurements hold the promise to improve high-precision sensing ranging from the definition of time standards to the determination of fundamental constants of nature. However, quantum sensors lose their sensitivity in the presence of noise. To protect them, the use of quantum error-correcting codes has been proposed. Trapped ions are an excellent technological platform for both quantum sensing and quantum error correction. Here we present a quantum error correction scheme that harnesses dissipation to stabilize a trapped-ion qubit. In our approach, always-on couplings to an engineered environment protect the qubit against spin-flips or phase-flips. Our dissipative error correction scheme operates in a continuous manner without the need to perform measurements or feedback operations. We show that the resulting enhanced coherence time translates into a significantly enhanced precision for quantum measurements. Our work constitutes a stepping stone towards the paradigm of self-correcting quantum information processing.

[1] Institute for Theoretical Physics, University of Innsbruck, A-6020 Innsbruck, Austria. [2] Institute for Quantum Optics and Quantum Information of the Austrian Academy of Sciences, A-6020 Innsbruck, Austria. [3] Harvard University, Department of Physics, Cambridge, MA 02138, USA. [4] Niels Bohr Institute, University of Copenhagen, Blegdamsvej 17, DK-2100 Copenhagen, Denmark. Correspondence and requests for materials should be addressed to F.R. (email: reiter@physics.harvard.edu)

**Q**uantum noise is a major obstacle for devices that take advantage of quantum mechanics, such as quantum computers[1], quantum networks[2], quantum simulators[3], and quantum-enhanced sensors[4–6]. The quest to find viable strategies for mitigating errors is thus an essential prerequisite for the development of quantum technologies and has led to techniques for quantum error correction[7–14]. Improving quantum sensing protocols in the presence of noise[15–21] by applying quantum error-correcting codes[22–26] represents a young research direction and impressive proof-of-concept realizations have already been demonstrated using nitrogen-vacancy centers[11–13]. Here we consider whether high-precision measurements can be improved by a self-correction mechanism induced by engineered dissipation. In particular we regard trapped-ion systems, which have proven to be an excellent platform for high precision measurements[27,28], as well as for the realization of quantum error-correcting codes[9,10].

Harnessing dissipative processes by engineering the coupling of a system to an environment or reservoir[29–37] provides a route for processing quantum information alternative to relying on unitary gate operations[9–14]. Such reservoir engineering techniques have for example been successfully applied for state preparation[38–42]. Dissipative schemes for preparing entangled resource states have been shown to have advantages over standard methods, leading for example to a better use of resources[34] and extending entanglement lifetimes by stabilizing the target state[38].

Employing dissipation for quantum error correction takes this idea further since it requires the stabilization of an unknown state (i.e., of a manifold of states). The idea of dissipative error correction has attracted considerable interest[43–50]. The challenge of implementing this strategy by engineering suitable dissipative processes in concrete experimental systems has recently led to theoretical proposals for superconducting circuits[51–55], as well as first experimental efforts towards the realization of building blocks required for dissipative quantum error correction[56]. Despite their central roles in both quantum information processing and quantum metrology, such achievements have not been made with trapped ions.

In this work, we address this challenge by combining the paradigms of dissipative quantum error correction and trapped-ion quantum information processing to a scheme for quantum error-corrected metrology. Standard error-correcting schemes entail classical apparatuses to perform measurements and feedback operations on the quantum system. In contrast to this gate-based approach, our scheme neither relies on time-dependent unitary operations, nor requires macroscopic measurements or feedback operations. Instead, tailored dissipative dynamics continuously corrects for spin-flips or phase-flips at a microscopic level by coupling the internal degrees of freedom of a system of trapped ions to an environment consisting of cooled motional modes (Fig. 1). The resulting dynamical protection of a qubit against noise results in a significant enhancement of its lifetime, and hence in a substantial improvement of quantum measurements. The proposed scheme allows for the realization of a repetition code[7], where a logical qubit is encoded in a three-particle entangled state. Dissipative processes are designed such that the code space is a steady-state dark manifold, as shown in Fig. 1. Errors take the quantum state out of this subspace, which causes engineered dissipative processes to become resonant and thereby coherently correct the error by a generalized optical pumping process. Once the error is corrected, the engineered dissipative processes are shifted out of resonance such that they cease to act on the system. As we show below, this allows one to stabilize a qubit against a given type of error: either single-qubit spin-flips or phase-flips, or against correlated noise. Adding to the potential of this approach for quantum information processing we demonstrate that the proposed scheme can be applied for improving the sensitivity of quantum sensing protocols, where a "dissipative" quantum-error correction paradigm has not been explored up to now. Here our scheme provides a blueprint for quantum error correction enhanced metrology in trapped-ion systems based on current experimental means. We analyze the applicability of our scheme in the context of a paradigmatic measurement setting with trapped ions and show that the attainable sensitivity can be significantly enhanced for realistic experimental parameters. Our work provides a stepping stone towards a new paradigm of self-correcting quantum systems that can be realized with current technology and will enable experiments with ions that take dissipative quantum information processing to a new level.

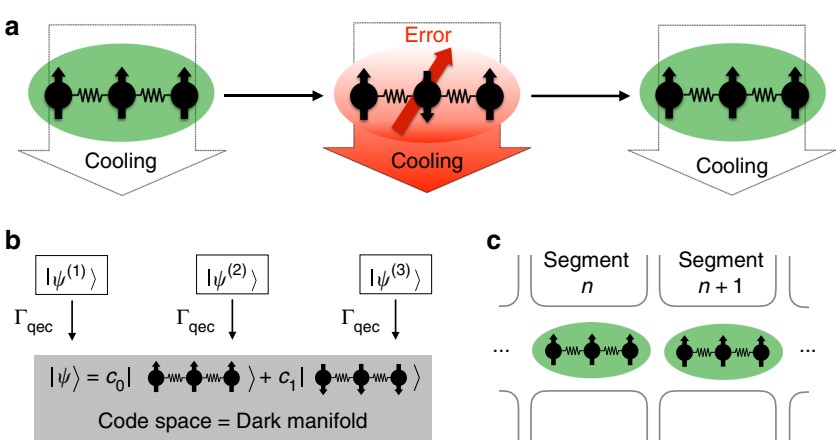

**Fig. 1** Dissipative realization of a three-qubit code in trapped ions. The unknown state $|\psi\rangle = c_0|000\rangle + c_1|111\rangle$ is encoded in the internal states of three ions and protected against single spin-flips by means of engineered couplings to a quantum reservoir that consists of motional modes. **a** Spin-flip errors are continuously removed from the system through cooling of the motional modes. **b** Compromised states $|\psi^{(j)}\rangle = \sigma_x^{(j)}|\psi\rangle$, $j = 1, 2, 3$ are driven back to the logical subspace at a rate $\Gamma_{qec}$, while the code space is a dark manifold of the engineered dissipative processes. **c** The scheme can be scaled up to several logical qubits using a segmented trap (Supplementary Note 3)

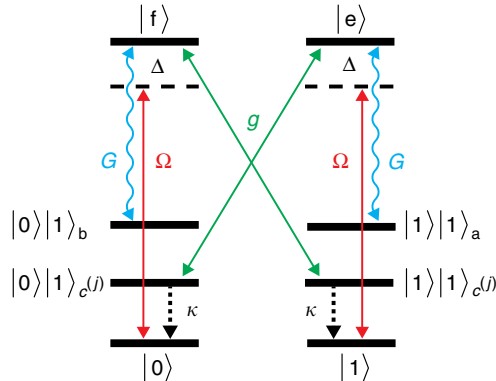

**Fig. 2** Setup for the implementation of the error correction scheme. The setup consists of three system ions, which couple collectively to two shared motional modes a and b, used for the interrogation of the system. For the correction, we require auxiliary modes $c^{(j)}$, which can be realized either by additional motional modes or ancilla ions. In the figure this additional degree of freedom is denoted by a level $|1\rangle_{c^{(j)}}$. The ions are assumed to have two levels, $|0\rangle$ and $|1\rangle$, which we shall refer to as ground states, and two levels, $|e\rangle$ and $|f\rangle$, referred to as excited states. The transitions are excited by a driving (weak carrier) field $\Omega$ and strongly coupled to two motional sidebands a and b with coupling constant G. Excitations of the ions ($|e\rangle$, $|f\rangle$) are coherently transferred to the auxiliary modes/ancillas by coherent couplings g and subsequently removed by cooling/reset with a rate $\kappa$

## Results

**Error-correcting protocol**. We consider a three-qubit repetition code, where a logical qubit is encoded in three physical qubits,

$$|\psi\rangle = c_0|0\rangle_L + c_1|1\rangle_L = c_0|000\rangle + c_1|111\rangle, \quad (1)$$

with logical qubit states $|0\rangle_L = |000\rangle$ and $|1\rangle_L = |111\rangle$, forming a codespace $\{|0\rangle_L, |1\rangle_L\}$. This encoding allows one to protect the quantum state against single spin-flips on the physical qubits, which lead to the single-error states

$$|\psi^{(j)}\rangle = \sigma_x^{(j)}|\psi\rangle, \quad j = 1, 2, 3, \quad (2)$$

where the Pauli operator $\sigma_x^{(j)} = |0\rangle_j\langle 1| + |1\rangle_j\langle 0|$ acts on the $j$th qubit. As explained below, a majority vote allows for the restoration of the state $|\psi\rangle$. Later we will also consider the correction of single-qubit phase-flips and correlated noise. In principle it is also possible to convert this scheme into a protocol simultaneously correcting spin-flip and phase errors by extending it to a nine-qubit error-correcting code[7] capable of correcting any type of single-qubit error. However, a full error-correcting code is not compatible with quantum sensing, since it would remove both errors and the signal to be measured from the dynamics.

In the following, we describe the noisy dynamics of a quantum system by a master equation $\dot{\rho} = \mathcal{L}(\rho)$ with a Liouvillian $\mathcal{L}$ of Lindblad form,

$$\mathcal{L}_{\text{noise}}(\rho) = \sum_k \mathcal{D}[L_k](\rho), \quad (3)$$

$$\mathcal{D}[L_k](\rho) = L_k\rho L_k^\dagger - \frac{1}{2}\left(L_k^\dagger L_k\rho + \rho L_k^\dagger L_k\right),$$

with dissipators $\mathcal{D}[L_k]$ and jump operators $L_k$. In the special case when $L_k$ describe local spin-flip noise affecting each of the constituent qubits independently, the jump operators are given by

$$L_x^{(j)} = \sqrt{\Gamma}\sigma_x^{(j)}, \quad j = 1, 2, 3, \quad (4)$$

where $\Gamma$ is the rate at which a spin flip occurs.

Single spin-flip errors are corrected by implementing a majority vote: to correct errors on, for example, the second qubit, we interrogate the two-body stabilizer operators $S^{(12)} = \sigma_z^{(1)}\sigma_z^{(2)}$ and $S^{(23)} = \sigma_z^{(2)}\sigma_z^{(3)}$, which involve the second qubit and its neighbors, qubit 1 and qubit 3. The code state $|\psi\rangle$ is an eigenstate of these operators with eigenvalue +1, while the single

error state $|\psi^{(2)}\rangle$ is an eigenstate with eigenvalue −1. If a spin-flip has occurred, the resulting state $|\psi^{(2)}\rangle$ of the system thus violates both stabilizers, $S^{(12)}|\psi^{(2)}\rangle = S^{(23)}|\psi^{(2)}\rangle = -|\psi^{(2)}\rangle$. Conditioned on this result, a spin-flip $\sigma_x^{(2)}$ is applied to qubit 2, and the original state $|\psi\rangle$ is restored. Errors on the first and third qubits are corrected in an analogous fashion. This recovery protocol can be implemented in a continuous manner by realizing the dissipative dynamics

$$\mathcal{L}_{\text{qec}} = \mathcal{D}[L_{x,\text{qec}}^{(1)}] + \mathcal{D}[L_{x,\text{qec}}^{(2)}] + \mathcal{D}[L_{x,\text{qec}}^{(3)}], \quad (5)$$

with quantum error-correcting jump operators of the form

$$L_{x,\text{qec}}^{(2)} = \sqrt{\Gamma_{\text{qec}}}\sigma_x^{(2)}\frac{\mathbb{1}-\sigma_z^{(1)}\sigma_z^{(2)}}{2}\frac{\mathbb{1}-\sigma_z^{(2)}\sigma_z^{(3)}}{2}. \quad (6)$$

Here, we have $\Gamma_{\text{qec}}$ as the correction rate, $\sigma_x^{(2)}$ as the correcting spin-flip, and two "interrogation parts" of the form $(\mathbb{1}-S)/2$. Each of them interrogates a stabilizer $S$, yielding 0 if the qubits are of the same value and $\mathbb{1}$ if they are different. This realizes the described majority vote: If both stabilizers $S^{(12)}$ and $S^{(23)}$ are violated, an action $L_{x,\text{qec}}^{(2)} \sim \sigma_x^{(2)}$ is realized. Correcting operators for qubits 1 and 3 can be written analogously:

$$L_{x,\text{qec}}^{(1)} = \sqrt{\Gamma_{\text{qec}}}\sigma_x^{(1)}\frac{\mathbb{1}-\sigma_z^{(1)}\sigma_z^{(2)}}{2}\frac{\mathbb{1}-\sigma_z^{(1)}\sigma_z^{(3)}}{2}, \quad (7)$$

$$L_{x,\text{qec}}^{(3)} = \sqrt{\Gamma_{\text{qec}}}\sigma_x^{(3)}\frac{\mathbb{1}-\sigma_z^{(1)}\sigma_z^{(3)}}{2}\frac{\mathbb{1}-\sigma_z^{(2)}\sigma_z^{(3)}}{2}. \quad (8)$$

For the physical implementation, it will be useful to translate the conditional jump operators in Eqs. (6)–(8) into the form

$$L_{x,\text{qec}}^{(j)} = \sqrt{\Gamma_{\text{qec}}}\left(\sigma_-^{(j)}P_{n_1=1} + \sigma_+^{(j)}P_{n_0=1}\right), \quad j = 1, 2, 3, \quad (9)$$

where $\sigma_+^{(j)}$ ($\sigma_-^{(j)}$) are the raising (lowering) operators on qubit $j$. $P_{n_k=n}$ are projectors on the states with $n$ qubits in state $|k\rangle$, e.g., $P_{n_1=1} = |100\rangle\langle 100| + |010\rangle\langle 001| + |001\rangle\langle 001|$ and $P_{n_0=1} = |011\rangle\langle 011| + |101\rangle\langle 101| + |110\rangle\langle 110|$. Similar conditional jump operators containing an interrogation part $(\mathbb{1}-S)/2$ were proposed in ref. [31] and implemented with trapped ions in ref. [39] using a sequence of quantum gates to dissipatively generate Bell pairs. In the following, we show how the conditional jump

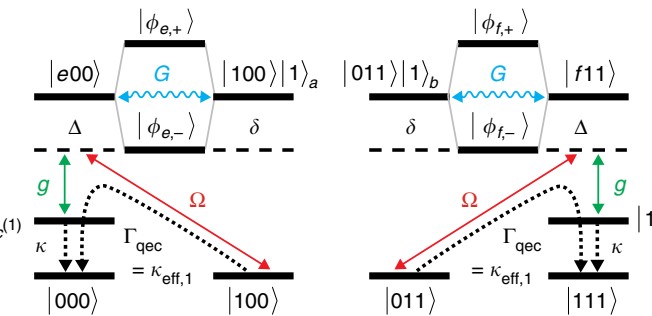

**Fig. 3** Error correction mechanism. The states after a spin-flip on ion 1, |100⟩ and |011⟩, are off-resonantly excited to the excited states |e00⟩ (left) and |f11⟩ (right). The strong sideband coupling $G$ results in the formation of the dressed states $|\phi_{e/f,\pm}\rangle$. For $\Delta = \delta = G$, the lower dressed states $|\phi_{e/f,-}\rangle$ are in resonance with the drives from the single-error states, allowing for their selective excitation. The excitations in |e⟩ and |f⟩ are transferred from the ion to the auxiliary mode $c^{(1)}$ by a coherent coupling, $g$, and removed by cooling, $\kappa$, maintaining the coherence between the two paths |100⟩ → |000⟩ and |011⟩ → |111⟩. Correction of spin-flips on other ions $j$ is performed in the same way, utilizing the auxiliary mode $c^{(j)}$

operator performing a majority vote in Eq. (9) can be implemented in a time-continuous fashion in a system of trapped ions.

The presented scheme can be generalized to correct for correlated spin-flip or phase errors, as explained in the "Methods" section.

**Setup**. We consider a setup consisting of three ions in a trap with identical levels and couplings as shown in Fig. 2. The logical qubit states |0⟩ and |1⟩ are encoded in internal electronic levels of the ions, which we shall refer to as ground states. In addition, we consider two levels, |e⟩ and |f⟩, referred to as excited states. The internal levels are coherently coupled to two collective motional modes of the ion chain, a and b, by applying appropriate laser fields. These modes are used for the interrogation of the system. In addition, for the correction of physical errors we require auxiliary modes $c^{(j)}$, which we discuss further down. The four internal states are assumed to be at least meta-stable and the modes are assumed to be cooled to the ground state. In a suitable rotating frame, the free Hamiltonian of the system is given by

$$H_{\mathrm{free},(\Delta,\delta)} = \Delta \sum_{j=1}^{3} \left( |e\rangle_j \langle e| + |f\rangle_j \langle f| \right) + \delta \left( a^\dagger a + b^\dagger b \right). \quad (10)$$

As will be explained below, the selectivity of the correction processes (i.e., spin-flips $\sigma_x^{(j)}$) to the single error states $|\psi^{(j)}\rangle$ will be achieved by a suitable choice of the detunings $\Delta$ and $\delta$. To interrogate the system, we apply weak carrier drives on the transitions |1⟩ ↔ |e⟩ and |0⟩ ↔ |f⟩,

$$H_{\mathrm{drive},\Omega} = \frac{\Omega}{2} \sum_{j=1}^{N} \left( |e\rangle_j \langle 1| + |f\rangle_j \langle 0| \right) + \mathrm{H.c.}, \quad (11)$$

and use sideband couplings that couple the transition |e⟩ → |1⟩ to interrogation mode a and the transition |f⟩ → |0⟩ to interrogation mode b,

$$H_{\mathrm{int},G} = G \sum_{j=1}^{3} \left( a^\dagger |1\rangle_j \langle e| + b^\dagger |0\rangle_j \langle f| \right) + \mathrm{H.c.} \quad (12)$$

This description is valid within the Lamb–Dicke approximation. The Lamb–Dicke approximation holds for $\eta^2(2\bar{n} + 1) \ll 1$, where $\bar{n}$ is the average phonon number and $\eta = 2\pi x_0/\lambda$ is the Lamb–Dicke parameter given by the ratio of the ion's ground state wave packet size $x_0$ and the wavelength of the applied light field $\lambda$. For the considered type of experiment, we consider the Lamb–Dicke parameter to be small. The coupling strength $G$ is assumed to be strong compared to all other rates in the system. The combination of the couplings $H_{\mathrm{drive},\Omega}$ and $H_{\mathrm{int},G}$, and the detunings in $H_{\mathrm{free},(\Delta,\delta)}$ will be used to identify ions which, after an error, reside in the state |1⟩ (|0⟩) and drive them to the state |e⟩ (|f⟩). While the interrogation modes a and b are needed to determine whether an error has occurred on the system ions, we also require couplings to additional auxiliary modes to remove such errors:

As can be seen from Eq. (9), the recovery process consists of two parts, one of which corrects states with a single qubit in state |1⟩, represented by $P_{n_1=1}$, by a lowering operation $\sigma_{-}^{(j)}$, e.g., |001⟩ → |000⟩. The second part corrects states with a single qubit in state |0⟩, i.e., $P_{n_0=1}$, by a raising operation $\sigma_{+}^{(j)}$, e.g., |110⟩ → |111⟩. For implementing the operation in Eq. (9) it is important to maintain the coherence between these two parts of the error correction. Thus, individual uncorrelated dissipation such as decay of |e⟩ and |f⟩ by spontaneous emission does not suffice. Instead, we use an engineered cooling process which is a combination of a mapping of the errors from the system ions to the auxiliary modes and a subsequent dissipative process to remove the errors as explained below. These auxiliary modes can be either additional ions or additional motional modes. The exact level structure is not important, the only requirement is that the first excited level needs to be strongly damped, thus effectively resulting in a two-level system. In the following, we focus on implementations based on the use of additional motional modes subjected to sympathetic cooling[40]. However, the analysis that follows can be straightforwardly adopted to the case of ancilla ions by replacing the bosonic operator $c^{(j)}$ in Eq. (13) and Eq. (15) by the Pauli operator $\sigma_{-}^{(j)}$. In this case, the required dissipative mechanism $L^{(j)} = \sqrt{\kappa}\sigma_{-}^{(j)}$ acting on each ancilla ion can be realized by continuously resetting its state by means of optical pump fields[39].

In the following we assume that for the correction of spin-flips acting independently on qubits 1, 2, and 3, the excitations of the system ions are mapped onto auxiliary motional modes $c^{(j)}$, $j = 1$, 2, 3. This is achieved by sideband couplings

$$H_{\mathrm{aux},g} = g \sum_{j=1}^{3} e^{i\delta_c t} (c^{(j)})^\dagger \left( |0\rangle_j \langle e| + |1\rangle_j \langle f| \right) + \mathrm{H.c.}, \quad (13)$$

with the coupling strength $g$ and the detuning $\delta_c$. The Hamiltonian of the overall system is given by

$$H_{\mathrm{total}} = H_{\mathrm{free},(\Delta,\delta)} + H_{\mathrm{drive},\Omega} + H_{\mathrm{int},G} + H_{\mathrm{aux},g}. \quad (14)$$

The resulting excitations in the motional modes are removed by a dissipative process,

$$L_c^{(j)} = \sqrt{\kappa} c^{(j)}, \quad (15)$$

for which we assume a large cooling rate $\kappa \gg g, \delta_c$ so that we can adiabatically eliminate $c^{(j)}$ from the dynamics[57]. This yields the engineered cooling operators

$$L_{\mathrm{eng}}^{(j)} = \sqrt{\kappa_{\mathrm{eng}}} \left( |0\rangle_j \langle e| + |1\rangle_j \langle f| \right), \quad (16)$$

with an engineered cooling rate $\kappa_{\mathrm{eng}} = g^2/\kappa$. The minor shift of the

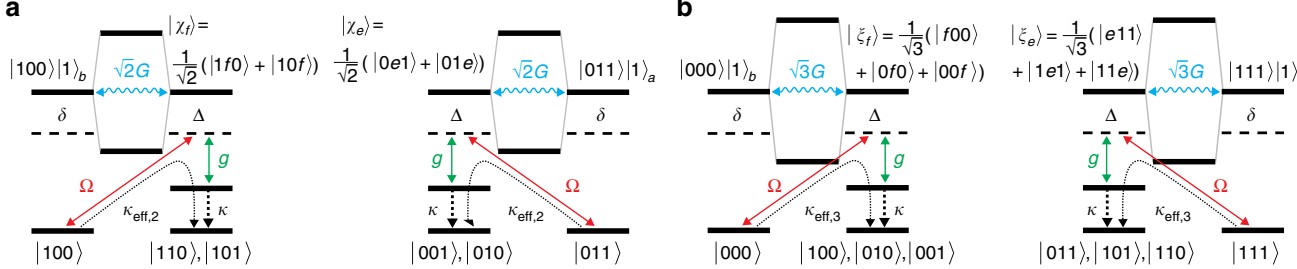

**Fig. 4** Intrinsic error processes. **a** Excitation of erroneous but correctable states, e.g., |100⟩ and |011⟩ can lead to excited states such as $|\chi_f\rangle = (|1f0\rangle + |10f\rangle)/\sqrt{2}$ instead of |e00⟩ and $|\chi_e\rangle = (|0e1\rangle + |01e\rangle)/\sqrt{2}$ instead of |f11⟩. Their dressed states with the motionally excited states |100⟩|1⟩$_b$ and |011⟩|1⟩$_a$ reside at detunings $|\Delta| = \sqrt{2}G$. For $\Delta = \delta = G$, such processes are off-resonant with respect to the drive, resulting in weak excitation. These processes, however, lead to a leakage of population from the subspace of correctable states since they generate double errors, e.g., |100⟩ → (|110⟩ + |101⟩)/√2 and |011⟩ → (|010⟩ + |001⟩)/√2, which cannot be corrected by our protocol. **b** Excitation of the logical states |000⟩ and |111⟩ leads to excited states |ξ$_f$⟩ and |ξ$_e$⟩ that couple to motionally excited states with strength $\sqrt{3}G$ and thus form dressed states at energies $\pm\sqrt{3}G$. These states are driven off-resonantly by the weak drive, which has a detuning $|\Delta| = G$. Such off-resonant excitations and subsequent engineered cooling processes transfer code states to the manifold with one error and can thus be recovered by the correction mechanism in the same manner as the physical error

levels by $H_{\text{aux},g}$ can be neglected and the Hamiltonian becomes

$$H_{\text{total,eng}} = H_{\text{free},(\Delta,\delta)} + H_{\text{drive},\Omega} + H_{\text{int},G}. \qquad (17)$$

As opposed to typical sources of noise such as spontaneous emission and decay of the vibrational modes, the engineered decay in Eq. (16) preserves the coherence between |e⟩ and |f⟩ ($c_0|e\rangle + c_1|f\rangle \to c_0|0\rangle + c_1|1\rangle$), which is a prerequisite for quantum error correction and for using the scheme in a quantum metrology setting as discussed below.

**Working mechanism**. We use the above couplings to implement the error-correcting dynamics in Eq. (9). To this end, we tailor effective jump processes mediated by engineered resonances involving the couplings to the excited levels and the motional modes in the setup. We carefully adjust the system parameters such that these dissipative mechanisms become resonant if the system is in a single-error state, resulting in a dynamical correction of the error. Detrimental action on the codespace and on the single-error space will be off-resonant and thus largely suppressed. Still, the remaining undesired dynamics introduce additional decay from the codespace and the single-error space towards higher-error spaces. Their rates need to be balanced by the choice of the available parameters, i.e., the strength of the drive and the engineered decay to ensure that these processes are not too detrimental.

Figure 3 illustrates the correction of the encoded qubit after a spin-flip on qubit 1. Spin-flips on qubits 2 and 3 are corrected analogously. Starting from the single-error state after a spin-flip on qubit 1 (Eq. (2)),

$$|\psi^{(1)}\rangle = c_0|100\rangle + c_1|011\rangle, \qquad (18)$$

the weak drive $H_{\text{drive},\Omega}$ couples |100⟩ to |e00⟩ and |011⟩ to |f11⟩. This excitation is a priori off-resonant due to the detunings in $H_{\text{free},(\Delta,\delta)}$. We now use the coupling $H_{\text{int},G}$ to create resonances in the excited state manifold which are selective to the number of qubits in |0⟩ or |1⟩ in the initial state: $H_{\text{int},G}$ couples |e00⟩ back to |100⟩ by exciting the interrogation mode a, resulting in the state |100⟩|1⟩$_a$. This coupling between |e00⟩ and |100⟩|1⟩$_a$ leads to the formation of dressed states,

$$\left|\phi_{e,\pm}\right\rangle = c_{e,\pm}|e00\rangle \pm c_{a,\pm}|100\rangle|1\rangle_a, \qquad (19)$$

which are indicated in Fig. 3. For $\Delta = \delta = G$, the lower dressed state resides at the detuning $\Delta_{1,-} = 0$ (the upper one at $\Delta_{1,+} = 2G$),

such that it is in resonance with the drive $H_{\text{drive},\Omega}$ from |100⟩. Hence, |100⟩ is excited to $|\phi_{e,-}\rangle$.

Simultaneously, the |011⟩-part in Eq. (18) is excited to the state |f11⟩. The excitation |f⟩ is coupled to interrogation mode b by $H_{\text{int},G}$, which results in the formation of the dressed states

$$\left|\phi_{f,\pm}\right\rangle = c_{f,\pm}|f11\rangle \pm c_{b,\pm}|011\rangle|1\rangle_b, \qquad (20)$$

with the energies $\Delta_{1,\pm}$. For the parameter choice $\Delta = \delta = G$, |011⟩ is resonantly excited to $|\phi_{f,-}\rangle$. Thus the state in Eq. (18) is excited to

$$|\phi_1\rangle = c_0|\phi_{e,-}\rangle + c_1|\phi_{f,-}\rangle. \qquad (21)$$

From here, excitation exchange with the auxiliary mode $c^{(1)}$ by $H_{\text{aux},g}$ transfers the system to $|\psi\rangle|1\rangle_{c^{(1)}}$. Cooling of the auxiliary mode by $L_c^{(1)}$ recovers the original state $|\psi\rangle$. The last two steps can be described as an effective decay process from $|\phi_1\rangle$ to $|\psi\rangle$ (cf. Eq. (16)), which, together with the coherent drive from $|\phi_1\rangle$ to $|\psi^{(1)}\rangle$, realizes the desired error correction on qubit 1. Errors on other qubits are corrected in an analogous fashion, utilizing the auxiliary mode $c^{(j)}$. We will later verify that this procedure indeed realizes the desired operators (9).

We remark that the correction of several types of errors can also be realized sequentially rather than simultaneously. In this case only a single auxiliary mode c is required. For correcting local spin-flips, individual time slots $T_1$, $T_2$, $T_3$ can be assigned for correcting spin-flip errors on qubit 1, qubit 2, and qubit 3 such that a sequence repeating these dedicated time slots $T_1$, $T_2$, $T_3$, $T_1$, $T_2$, $T_3$, $T_1$... corrects the errors on the individual qubits one after the other. Such a "Trotterization" would, however, lead to an effective decrease in the correction rate by a factor of three. The error rates would, on the other hand, be unchanged such that the increase in the lifetime of the codeword would be less pronounced than in the continuous case.

Apart from the error-correcting mechanism, the scheme also entails undesired processes where spins are not flipped in accordance with the majority vote. As illustrated in Fig. 4, this includes processes such as |100⟩ → |110⟩ or |000⟩ → |100⟩. For example, the undesired excitations from |100⟩ to the $f$-excited state $|\chi_f\rangle = (|1f0\rangle + |10f\rangle)/\sqrt{2}$ and from |011⟩ to the $e$-excited state $|\chi_e\rangle = (|0e1\rangle + |01e\rangle)/\sqrt{2}$, shown in Fig. 4a), are suppressed by our parameter choice of $\Delta = \delta = G$. This can be understood as follows: $H_{\text{int},G}$ couples $|\chi_f\rangle$ to |100⟩|1⟩$_b$. Due to constructive interference between the two terms in $|\chi_f\rangle$, this coupling has a strength of $\sqrt{2}G$. The resulting dressed states thus

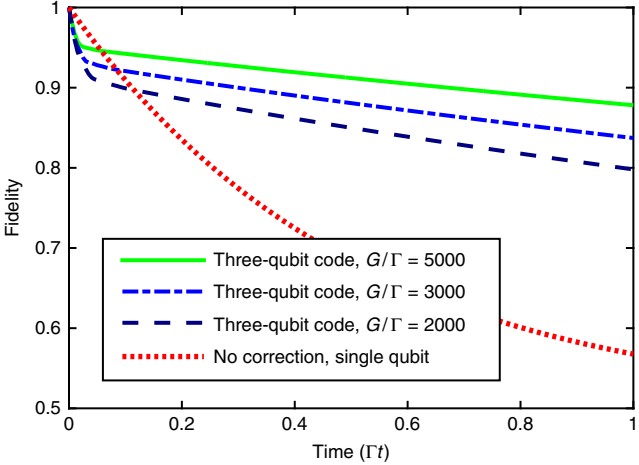

**Fig. 5** Continuous error correction of a three-qubit code against spin-flip noise compared to single-qubit decay. We simulate the dynamics of a system of three physical qubits subjected to individual spin-flip errors at a rate $\Gamma$ under the action of the error correction scheme by simulating the full master equation $\dot{\rho}(t)$ restricted to at most one excitation. The shown results are obtained from optimizing the fidelity (see main text) at $t = 1/\Gamma$ by the choice of $\kappa_{\mathrm{eng}}$ and $\Omega$ for each considered value of the sideband coupling $G$. For comparison, the red dotted line displays the decay of a single qubit subject to spin-flips at a rate $\Gamma$. We find that the implementation of our scheme in trapped ions leads to a significantly reduced decay: Assuming a sideband coupling of $G = 5000\Gamma$ (green solid line), a fidelity of $F \approx 0.9$ of the dissipatively protected logical qubit (encoded in three decaying physical qubits) is maintained after a time $t \sim 1/\Gamma$

reside at energies $\Delta_{2,\pm} \approx (1 \pm \sqrt{2})G$ such that neither of the dressed states are in resonance with the drive. As a consequence, these processes taking the system away from the desired state will be much slower than the resonant processes correcting the errors. Excitation from $|011\rangle$ to $|\chi_e\rangle$ is suppressed for the same reason. The same arguments hold for states after single errors on other qubits. The resulting slow process leads to a loss of population from the subspace of single-error states to the subspace of double-error states, which we discuss below.

In addition, the operation of the protocol also causes losses from the logical subspace, as shown in Fig. 4b): $|000\rangle$ is excited to $|\xi_f\rangle = (|f00\rangle + |0f0\rangle + |00f\rangle)/\sqrt{3}$ and $|111\rangle$ to $|\xi_e\rangle = (|e11\rangle + |1e1\rangle + |11e\rangle)/\sqrt{3}$. These superposition states consist of three terms and couple to $|000\rangle|1\rangle_b$ and $|111\rangle|1\rangle_a$ with a coupling strength $\sqrt{3}G$. The energies of the resulting dressed states are therefore given by $\Delta_{3,\pm} = (1 \pm \sqrt{3})G$. Hence, this excitation is also off-resonant with respect to the drive, leading only to a weak additional error process, which is, however, intrinsic to the scheme.

To optimize the performance of the error correction in the presence of these undesired dynamics, we invoke a quantitative model of the effective dynamics of the system further down. We find that the lifetime of the codeword is maximized by balancing the intrinsic loss rates through the choice of the engineered decay rate $\kappa_{\mathrm{eng}}$ and the driving strength $\Omega$. Optimizing for long operation times, we find that the optimum occurs when the total decay rates (i) from the codespace to the single-error space and (ii) from the single-error space to the double-error space are small and similar in magnitude. Still, the precise values of the couplings $\Omega$, $G$, $g$, and $\kappa$ are not critical in our scheme. To maintain the coherence of the codeword $|\psi\rangle = c_0|000\rangle + c_1|111\rangle$ under the correction in Eq. (9), we require that the rate $\Gamma_{\mathrm{qec}}$ is the same for both parts of the superposition, which is for example fulfilled if $\Omega$, $G$, and $g$ are identical for both paths illustrated in Fig. 3. To

achieve a maximum correction rate, the detunings $\Delta$ and $\delta$ need to be tuned with an accuracy of at least $\kappa_{\mathrm{eng}}$.

**Effective model.** In order to analytically verify that our scheme realizes the desired operators, we use the effective operator formalism[57] to adiabatically eliminate the excited degrees of freedom, which are coupled to the stable ground states by perturbative coherent couplings and are only weakly populated. Adiabatic elimination of the internal levels $|e\rangle$ and $|f\rangle$ (see "Methods") leads to effective ground state dynamics (involving only the internal levels $|0\rangle$ and $|1\rangle$ with motional modes in the ground state) that is described by an effective master equation with jump operators

$$L_{\mathrm{eff}}^{(j)} = \sum_{n=1}^{3} \sqrt{\kappa_{\mathrm{eff},n}} \left( \sigma_-^{(j)} P_{n_1=n} + \sigma_+^{(j)} P_{n_0=n} \right). \quad (22)$$

The above jump operators contain the desired error-correcting terms ($n = 1$) given by Eq. (9). These result in a decay of the subspace of single-error states to the logical manifold at a large effective rate $\Gamma_{\mathrm{qec}} \equiv \kappa_{\mathrm{eff},1} \approx \Omega^2/\kappa_{\mathrm{eng}}$, where $\kappa_{\mathrm{eng}} = g^2/\kappa$ is the engineered cooling rate introduced in Eq. (16). In addition to these error-correcting processes, we also obtain weak undesired decay terms (Fig. 4). These include (i) terms with $n = 3$ that act on the logical manifold introducing single-qubit errors (e.g., $|000\rangle \rightarrow |100\rangle$) at a rate $\kappa_{\mathrm{eff},3}$ and (ii) terms with $n = 2$, transferring single-error states to uncorrectable double-error states (e.g., $|110\rangle \rightarrow |100\rangle$). As discussed above, these processes are detuned by an amount $\sim G$ and are therefore strongly suppressed. We verify this explicitly in Supplementary Note 2.

**Performance of the scheme.** We analyze the dynamics of our protocol analytically and numerically. For the former, we use the error correction rate $\Gamma_{\mathrm{qec}} = \kappa_{\mathrm{eff},1}$ and the rates of undesired processes $\kappa_{\mathrm{eff},2}$ and $\kappa_{\mathrm{eff},3}$ to describe the effective dynamics of the scheme by a system of coupled rate equations (Supplementary Note 2). For the numerical analysis, we simulate the full dynamics of the system without using the effective operator formalism. This is modeled by a master equation $\dot{\rho} = \mathcal{L}_{\mathrm{total}}(\rho)$ with the Liouvillian $\mathcal{L}_{\mathrm{total}} = \mathcal{L}_{\mathrm{noise}} + \mathcal{L}_{\mathrm{qec}}$, which accounts for the noise processes we aim to correct for and the physical couplings of the error correction scheme (see "Methods"). The error-correcting code is assumed to operate on three physical qubits that are each subject to spin-flip errors acting at a rate $\Gamma$. We calculate the fidelity $F(t) = \mathrm{Tr}\{\rho(t)|\psi(0)\rangle\langle\psi(0)|\}$ with respect to the initial state $|\psi(0)\rangle = (|000\rangle + i|111\rangle)/\sqrt{2}$ and compare the result with the decay of a single qubit subject to spin-flips (cf. Eq. (4), with a rate $\Gamma$), that is initially prepared in the state $|\psi(0)\rangle = (|0\rangle + i|1\rangle)/\sqrt{2}$. For different sideband coupling strengths, we numerically optimize the parameters $\kappa_{\mathrm{eng}}$ and $\Omega$ to achieve maximum fidelity at the time $t = 1/\Gamma$. The number of excitations in the simulation is limited to at most one. Given that we mostly operate in a regime of weak driving $\Omega^2 \ll \kappa_{\mathrm{eng}}^2$, $G^2$ this constitutes a good approximation. Truncating at higher numbers of excitations is found not to result in a notable difference of the evolution.

The results are shown in Fig. 5, where we plot the dynamical evolution of the system for different values of $G/\Gamma$ (see Supplementary Figure 1 in Supplementary Note 2 for a wider range of sideband couplings). It can be seen that applying the dissipative three-qubit error correction code yields a significant advantage compared to using a single decaying qubit. For $G = 5000\Gamma$ the code maintains a fidelity close to 0.9 at $t \sim 1/\Gamma$, where the single-qubit fidelity has almost dropped to the steady state value of 0.5.

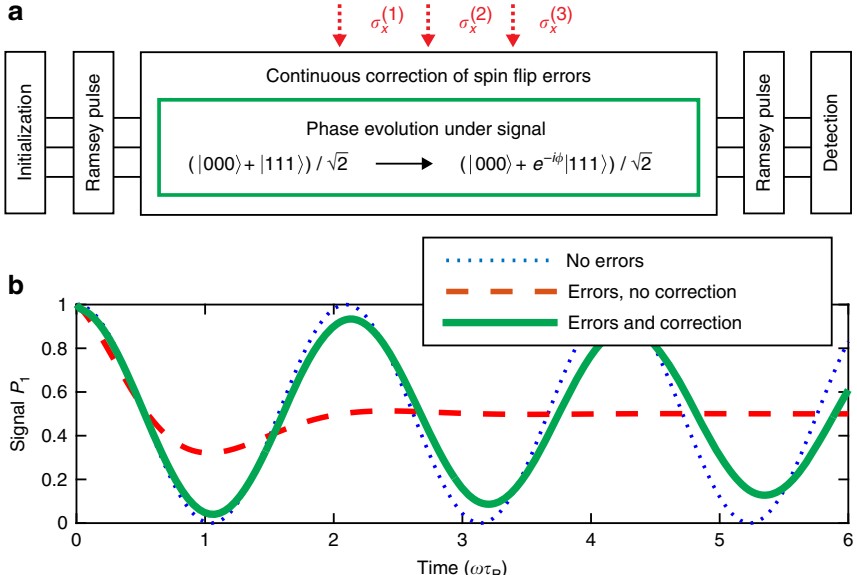

**Fig. 6** Application of the proposed error correction scheme to quantum metrology. **a** Ramsey scheme. From an initial state $|000\rangle$, a Ramsey pulse prepares $(|000\rangle + |111\rangle)/\sqrt{2}$. Under $H = (\omega/2)\sum_{j=1}^{N} \sigma_z^{(j)}$, the state evolves to $(|000\rangle + e^{-i\phi}|111\rangle)/\sqrt{2}$ during a Ramsey time $\tau_R$. Rotation by a second Ramsey pulse and measurement of the population $P_1$ of one ion in the $\{|0\rangle,|1\rangle\}$ basis allows deducing $\omega$ according to $P_1 = \cos^2(\phi/2)$ with $\phi = 3\omega\tau_R$. **b** In the absence of errors, the Ramsey measurement has full fringe contrast (dotted blue line). Spin-flips at a rate $\Gamma = \omega/2$ damp out the oscillations (dashed red line). Applying the presented scheme for error correction protects the evolution and restores the fringe contrast (solid green line, assuming a sideband coupling of $G = 5000\Gamma$)

The decrease in the fidelity of the codeword generally takes a hockey-stick-shaped form consisting of two parts: a fast initial drop and a slow exponential decay. This shape is a characteristic feature of time-continuous quantum error-correcting schemes (see, e.g., ref. [48]). The initial drop is due to the fact that physical decay processes acting on the three physical qubits and laser-induced error processes lead to a loss of population from the codespace. The losses set in immediately, while the error correction-induced transfer back to the codespace sets in with a delay that is determined by the timescale on which population accumulates in the single-error space and $\Gamma_{qec}$. The subsequent effective decay rate in the presence of error correction is substantially reduced as compared to the single-qubit decay rate. This is clearly visible in Fig. 5.

In Supplementary Note 2, we introduce a rate equation model, which allows us to quantitatively describe the dynamics and to derive the optimal parameters for the scheme. We find that the fidelity of the codeword initially drops to $F_0 = 1 - E_0$ with $E_0 \approx \Gamma_{err}/\Gamma_{qec}$, where $\Gamma_{err} = 3(\Gamma + \kappa_{eff,3})$ is the loss rate from of the logical subspace. For the slow exponential decay we obtain an effective rate $\Gamma_{eff} \approx \Gamma_{err}\Gamma_{leak}/\Gamma_{qec}$, where $\Gamma_{leak} = 2(\Gamma + \kappa_{eff,2})$ is the total leakage rate from the subspace of single-error states to the subspace of two-error states (Supplementary Note 2). Assuming perfect error correction ($G \to \infty$), the detrimental rates approach the bare spin-flip rates, thus $\Gamma_{err} \to 3\Gamma$, $\Gamma_{leak} \to 2\Gamma$, and the initial drop and effective decay rate become $E_0 \approx \Gamma/\Gamma_{qec}$ and $\Gamma_{eff} \approx 6\Gamma^2/\Gamma_{qec}$, as expected for a continuous implementation of the three-qubit code[43,48].

For finite sideband coupling $G$, we maximize the fidelity for long times by minimizing $\Gamma_{eff}$ (the details of the optimization are given in Supplementary Note 2). This leads to an optimal parameter choice $\Omega \approx \kappa_{eng} \approx 1.2(\Gamma G^2)^{1/3}$, which allows for a correction rate $\Gamma_{qec} \approx \kappa_{eng}/3$ and leads to small and similar error and leakage rates, $\Gamma_{err} \approx 3.3\Gamma$ and $\Gamma_{leak} \approx 2.7\Gamma$. The resulting initial drop and effective decay rate are $E_0 \approx 10(\Gamma/G)^{2/3}$ and $\Gamma_{eff} \approx 27(\Gamma/G)^{2/3}\Gamma$. Alternatively, the protocol can be optimized for short operation times by minimizing the initial drop $E_0$.

In "Methods" section, we discuss imperfections that can occur in realistic setups and can lead to a reduction of the fidelity of the logical state. We address the effect of (i) decoherence associated with the decay of the excited degrees of freedom, (ii) imperfect cooling and heating of the motional modes, and (iii) the effect of "complementary" errors that are not corrected by our scheme. More specifically, our scheme is designed to correct for one type of errors (for example spin-flip errors). Errors other than the targeted type cannot be corrected simultaneously in the present version of the protocol. The performance of a scheme that corrects spin-flips ($x$-errors), will for example be degraded by the presence of "complementary" $z$-errors.

**Application to quantum metrology.** Quantum error correction protocols find attractive applications for quantum metrology, as has been proposed for example in refs. [22–24]. Following these ideas, we explore the application of our error correction scheme for improving high-precision measurements of weak magnetic fields with ions. To this end, we study a prototypical setting, where a Ramsey-type protocol (Fig. 6a) is used to measure a signal originating from a field in the $z$-direction acting on several probe particles $H = (\omega/2)\sum_{j=1}^{N} \sigma_z^{(j)}$. We commence by considering three probe particles constituting one logical qubit. The measurement sequence involves four steps. (i) Starting from the initial state $|000\rangle$, a first Ramsey pulse (a $\pi/2$ rotation on the logical qubit) prepares the superposition state $|\psi(0)\rangle = (|000\rangle + |111\rangle)/\sqrt{2}$ using standard gate operations in ions[9,10]. (ii) During a Ramsey waiting time of duration $\tau_R$, the superposition state picks up a relative phase $\phi(\tau_R) = 3\omega\tau_R$, such that $|\psi(\tau_R)\rangle = (|000\rangle + e^{-i\phi(\tau_R)}|111\rangle)/\sqrt{2}$. (iii) A second Ramsey pulse (another $\pi/2$ rotation on the logical qubit) transforms the evolved state into $|\psi'(\tau_R)\rangle = \cos(\phi(\tau_R)/2)|111\rangle + i\sin(\phi(\tau_R)/2)|000\rangle$. (iv) Finally, a measurement in the $\sigma_z$-basis is performed on one of the qubits. The probability to detect the first (or any other) qubit in state $|1\rangle$ is given by $P_1 = \cos^2(\phi(\tau_R)/2)$, which allows one to infer the phase $\phi(\tau_R)$ and thus the signal strength $\omega$. As explained in "Methods", the sensitivity of this measurement is

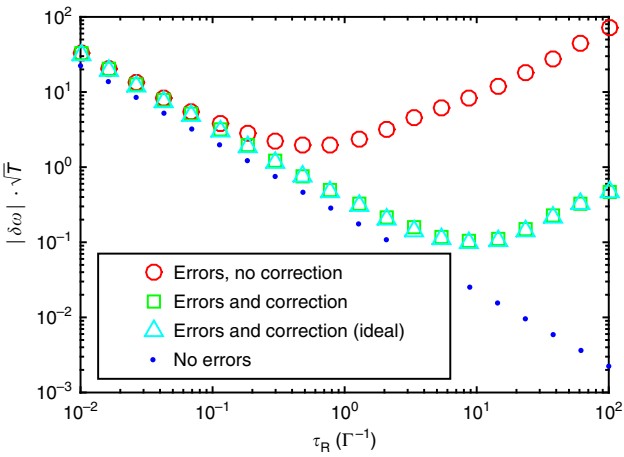

**Fig. 7** Quantum error correction enhanced sensing. The plot shows the normalized measurement precision $|\delta\omega|\sqrt{T}$ vs. the Ramsey time $\tau_R$ for a Ramsey measurement subject to spin-flips. The action of the error-correcting scheme can be seen to improve the achievable sensitivity by about an order of magnitude: The shown results display the normalized measurement precision assuming errors and (i) no error correction, (ii) error correction using the ideal jump operators given in Eq. (9), and (iii) error correction including undesired processes using the jump operators given in Eq. (22). As reference, the precision in the absence of errors (iv) is also shown. For the numerical simulation, we use $G = 5000\Gamma$, where $\Gamma$ is the spin-flip rate, and the optimized parameters $\Omega = 4\kappa_{\text{eng}}/5$ and $\kappa_{\text{eng}} = 1.2$ $(\Gamma G^2)^{1/3}$ discussed in the text, and restrict the full master equation to the subspace of at most one excitation

given by $|\delta\omega| = 1/(N\sqrt{\tau_R}\sqrt{T})$, where $N$ is the number of probe particles ($N = 3$ in our case) and $T$ is the total measurement time ($T = m\tau_R$, where $m$ is the number of runs). Increasing the Ramsey time leads to an improvement of the measurement precision (Fig. 7).

In the presence of noise, the precision of the measurement changes significantly[5,15,17–19], given that both the amplitude and period of the fringes (cf. Fig. 6b) are affected by the noise. It has been shown theoretically[26] that quantum error correction can be used to restore the optimal measurement precision in the presence of directed noise, as long as it is not parallel to the signal. Noise processes acting along the same direction as the signal ($z$) cannot be distinguished from the signal and thus cannot be targeted (we consider their effect in Supplementary Fig. 3). In the following, we analyze how well the detrimental effect of noise in the orthogonal direction ($x$) can be mitigated by the error correction protocol. Generally, if a signal in an arbitrary direction is assumed, correction of noise in $x$-direction as described below will allow for sensing the $z$-component of the signal with enhanced precision[26].

In the Ramsey-type sensing scheme above, we consider a signal in $z$-direction and transverse noise in the form of local spin-flips, $L_x^{(j)} = \sqrt{\Gamma}\sigma_x^{(j)}$. This degrades the state with a resulting decrease of the fringe contrast (as can be seen in Fig. 6b). As shown in Fig. 7, the normalized sensitivity $\sqrt{T}|\delta\omega|$ improves up to Ramsey waiting times $\tau_R \sim \Gamma^{-1}$, which limits the achievable measurement precision. The application of our quantum error correction scheme during step (ii) of the protocol (free evolution for a time $\tau_R$) thus reduces the speed at which the state is degraded. This shifts the optimum Ramsey waiting time to higher values, which improves the achievable measurement sensitivity. Figure 7 illustrates this effect for realistic experimental parameters and demonstrates that the presented error correction scheme

significantly improves the performance of the Ramsey sequence compared to the situation where transversal errors are not corrected. The achievable sensitivity in the presence of both transversal and parallel noise ($\sigma_x$ and $\sigma_z$ errors) is shown in Supplementary Fig. 3.

So far our discussion has involved one logical qubit. The scheme can be scaled up to involve $N_{\text{log}}$ logical qubits using a segmented trap[60,61], as shown in Fig. 1c) and explained in Supplementary Note 3. In this case, the potential barriers can be ramped up to divide the trap into segments confining three ions each. For sufficiently high barriers, the motional modes of the ion triples are independent from each other such that each logical qubit can be individually protected. This ramping into individual segments can be performed after the qubits are merged into an entangled state containing all qubits. Thereby the protocol can be used to obtain maximal quantum enhanced sensitivity while simultaneously being protected against spin-flip errors.

## Discussion

In summary, we have presented a continuous quantum error correction scheme for trapped ions that dynamically stabilizes an encoded qubit by coupling it to an engineered reservoir. This protocol does not require the use of a classical measurement apparatus or classical feedback loops. The error correction mechanism results from a built-in back-action mechanism induced by engineered dissipative processes. From a practical perspective, avoiding the measurement procedure is a major advantage since measurements typically require scattering thousands of photons, which is a slow process and heats up the motional mode. While our scheme relies on the coupling of internal degrees of freedom to motional modes that underpins the realization of quantum gates in trapped ions[9,10], it does not involve a sequence of gates, as in standard approaches to quantum error correction. Instead, our protocol uses the interaction between internal degrees of freedom and motional modes directly and requires only always-on couplings that act simultaneously and continuously on the encoded qubit. In this way, the role of auxiliary modes, to which the error syndromes are mapped, can be naturally played by the motional degrees of freedom, which allows one to continuously remove errors by means of standard sympathetic cooling of the motional modes.

The proposed error correction scheme can for example be used for quantum sensing, as we showed for the case of one logical qubit. By protecting random superposition states of three qubits we significantly increase the lifetime of the coherent oscillations in a Ramsey measurement. As a result, the sensitivity can be improved and the optimal measurement time can be shifted to higher values. For scaling up this approach, segmented ion traps can be used that can accommodate several logical qubits in different trap segments.

Our protocol pushes forward the boundary of dissipation-driven quantum information processing towards universal dissipative quantum computing. Engineered dissipation has already found useful applications for quantum state preparation, where the initial state is destroyed in the process of preparing a desired target state. In contrast, quantum error-correcting schemes need to preserve coherences in the initial state of a quantum system. Since such quantum error correction inherently relies on dissipation to get rid of entropy, it naturally fits into the framework of engineered dissipation. The realization of dissipative protocols that are capable of manipulating non-orthogonal quantum states while maintaining their coherence is an essential step in the endeavor to perform dynamically stabilized quantum information processing tasks. As the couplings assumed for the implementation of our scheme are generic couplings of a register of qubits to

motional modes/ancilla qubits, the presented working mechanism can be adopted to other systems such as superconducting architectures and Rydberg atoms.

The principles we have employed to implement a dissipative three-qubit code can be taken further towards more sophisticated quantum error-correcting codes. Here it will be interesting to tailor suitable many-body dissipation to protect larger classes of stabilizers with the aim to realize, for example, topological error correction. Moreover, it will be very interesting to develop codes that do not only correct for noise of Pauli type as discussed in this article, but also for other error types such as loss errors. Hybrid schemes, where the paradigm of dissipative quantum error correction is combined with the technique of monitoring environmental degrees of freedom, have already been proven to be useful for sensing[62] and may provide a promising route towards realizing such advanced schemes[38].

We remark that the dissipative confinement of a quantum system to a desired subspace could also be useful for quantum simulations of lattice gauge theories, where devising methods for limiting the dynamics of a system to the gauge-invariant part of the Hilbert space is a key challenge for the development of future quantum simulators[63,64]. From a broader perspective, the design of dissipative maps can find a wide range of applications for quantum information processing including dissipative schemes for entanglement distillation[65], generalized quantum measurements and the simulation of exotic quantum channels[55].

## Methods

**Dynamical model.** The dynamics of the system is modeled by a master equation $\dot{\rho} = \mathcal{L}_{\text{total}}(\rho)$ with the Liouvillian $\mathcal{L}_{\text{total}} = \mathcal{L}_{\text{noise}} + \mathcal{L}_{\text{qec}}$, where $\mathcal{L}_{\text{noise}}$ describes the noise processes we aim to correct for (Eq. (3)), and $\mathcal{L}_{\text{qec}}$ contains the physical couplings required for the error correction scheme,

$$\mathcal{L}_{\text{qec}}(\rho) = -i[H_{\text{total,eng}}, \rho] + \sum_{j=1}^{3} \mathcal{D}[L_{\text{eng}}^{(j)}](\rho).\qquad(23)$$

Here the Hamiltonian $H_{\text{total,eng}}$ is given by Eq. (17) and the jump operators $L_{\text{eng}}^{(j)}$ by Eq. (16). Simulation of the master equation allows us to verify the action of the proposed error-correcting scheme numerically.

**Effective dynamics of the open quantum system.** We reduce the full dynamics of the system to effective dynamics of the ground states by adiabatically eliminating the excited degrees of freedom. To this end, we use the effective operator formalism[57]. Here we assume that the stable ground states are coupled to the excited states by perturbative coherent couplings $V = V_{+} + V_{-}$, where $V_{+}$ ($V_{-}$) denotes (de-)excitation. The coherent and dissipative dynamics of the excited states is described by a non-Hermitian Hamiltonian

$$H_{\text{NH}} = H_e - \frac{i}{2}\sum_k L_k^\dagger L_k,\qquad(24)$$

where $H_e$ contains the coherent couplings between the excited states and $L_k$ are decay processes taking them to the ground states. Applying the formalism[57], we then obtain the effective operators

$$H_{\text{eff}} = -\frac{1}{2}V_{-}H_{\text{NH}}^{-1}V_{+} + \text{H.c.}\qquad(25)$$

$$L_{k,\text{eff}} = L_k H_{\text{NH}}^{-1}V_{+}.\qquad(26)$$

Here, $H_{\text{eff}}$ is the effective Hamiltonian and $L_{k,\text{eff}}$ are the effective jump operators. The resulting effective dynamics is described by an effective master equation

$$\dot{\rho} = -i[H_{\text{eff}}, \rho] + \sum_k \mathcal{D}[L_{k,\text{eff}}](\rho).\qquad(27)$$

From this reduced model we derive the rates for the error correction and leakage processes. Considering a simplified rate equation model allows us to assess the performance of the scheme and to derive and optimize the available parameters of the scheme analytically. The detailed calculations and a numerical comparison to the full dynamics of Eq. (23) can be found in Supplementary Note 2.

**Sensitivity of the measurement.** The sensitivity of a Ramsey spectroscopy scheme with entangled particles[5,66–68] is determined as follows. Using the Ramsey sequence described in the main text, a first Ramsey pulse prepares a system of $N$ qubits in the state $|\psi(0)\rangle^{\otimes N} = (|0\rangle^{\otimes N} + |1\rangle^{\otimes N})/\sqrt{2}$, which evolves under the Hamiltonian $H = (\omega/2)\sum_{j=1}^{N}\sigma_z^{(j)}$ for a time $\tau_R$, resulting in the state $|\psi(\tau_R)\rangle = (|0\rangle^{\otimes N} + e^{-iN\omega\tau_R}|1\rangle^{\otimes N})/\sqrt{2}$. After the second Ramsey pulse one of the qubits is measured. The probability to find this qubit in state $|1\rangle$ is given by $P_1 = \cos^2(N\omega\tau_R/2)$. The uncertainty in estimating $P_1$ due to the statistical fluctuations associated with a finite sample $\Delta P_1 = \sqrt{P_1(1 - P_1)/n_{\text{data}}}$, depends on the number of experimental data $n_{\text{data}}$. In the considered case this equals the number of runs $n_{\text{runs}} = n_{\text{data}} = T/\tau_R$, where $T$ is the total measurement time. The uncertainty in the measurement of $\omega$ is therefore given by

$$|\delta\omega| = \frac{\Delta P_1}{|dP_1/d\omega|}.$$

In the ideal case, this yields $|\delta\omega| = 1/(N\sqrt{T\tau_R})$. The numerical results in Figs. 6 and 7 have been obtained by calculating the time evolution of the density matrix of a three-ion system $\rho(t)$ in the presence of the signal Hamiltonian, individual spin-flips and the error-correcting scheme. Supplementary Figure 3 in Supplementary Note 3 shows the dynamics that is obtained if collective dephasing is added to the problem.

**Correction of other types of errors.** Our scheme can be generalized to correct for correlated spin-flip or phase errors, as detailed in Supplementary Note 1.

The action of correlated spin-flips, $L_x = L_x^{(1)} + L_x^{(2)} + L_x^{(3)} = \sqrt{\Gamma_X}\left(\sigma_x^{(1)} + \sigma_x^{(2)} + \sigma_x^{(3)}\right)$ leaves the system in a superposition state of all single-qubit spin-flip errors, $|\psi_X\rangle = (|\psi^{(1)}\rangle + |\psi^{(2)}\rangle + |\psi^{(3)}\rangle)/\sqrt{3}$. To correct for such an error, we replace the coupling of the ions to the three auxiliary modes $c^{(j)}$ in Eq. (13) by a coherent coupling to a single auxiliary mode $c$,

$$H_{\text{aux},g} = gc^\dagger \sum_{j=1}^{3} e^{i\delta_c t}\left(|0\rangle_j\langle e| + |1\rangle_j\langle f|\right) + \text{H.c.}\qquad(28)$$

The drive in Eq. (11) also coherently acts on the three ions. Correlated errors of all qubits are thereby coherently mapped onto c. Dissipating the excitations of c by $L_c = \sqrt{\kappa}c$ (cf. Eq. (15)) then realizes the single collective jump operator $L_{X,\text{qec}} = L_{x,\text{qec}}^{(1)} + L_{x,\text{qec}}^{(2)} + L_{x,\text{qec}}^{(3)}$, with $L_{x,\text{qec}}^{(j)}$ as in Eqs. (6)–(8).

To generalize the scheme to phase-flips, we perform the mapping $|0\rangle \to |+\rangle$ and $|1\rangle \to |-\rangle$, where $|\pm\rangle = (|0\rangle \pm |1\rangle)/\sqrt{2}$. The resulting codeword is given by $|\psi\rangle = c_+|+++\rangle + c_-|---\rangle$. Phase-flips on the second qubit are corrected by the action of the jump operator

$$L_{z,\text{qec}}^{(2)} = \sqrt{\Gamma_{\text{qec}}}\sigma_z^{(2)}\frac{\mathbb{1} - \sigma_x^{(1)}\sigma_x^{(2)}}{2}\frac{\mathbb{1} - \sigma_x^{(2)}\sigma_x^{(3)}}{2}.\qquad(29)$$

Errors on the first and third qubit are corrected analogously.

To correct for correlated phase-flips, we proceed as in the case of spin-flips above and adapt the scheme for individual errors to collective errors. We replace the couplings of the ions to individual auxiliary modes $c^{(j)}$ by a coherent coupling of all three ions to a single auxiliary mode $c$,

$$H_{\text{aux},g} = gc^\dagger \sum_{j=1}^{3} e^{i\delta_c t}\left(|+\rangle_j\langle e| + |-\rangle_j\langle f|\right) + \text{H.c.},\qquad(30)$$

Phase errors of the qubits are thus coherently mapped on the auxiliary mode c. Subsequent cooling of $c$ (Eq. (15)) then yields the jump operator $L_{Z,\text{qec}} = L_{z,\text{qec}}^{(1)} + L_{z,\text{qec}}^{(2)} + L_{z,\text{qec}}^{(3)}$ which combines the individual operators $L_{z,\text{qec}}^{(j)}$ for $j = 1, 2, 3$ (cf. Eq. (29) for $j = 2$) and thus corrects for correlated phase errors.

**External imperfections.** In the following, we discuss imperfections that can occur in realistic setups and may lead to a reduction of the fidelity of the logical state.

Our scheme is designed to correct for one type of individual or collective errors (for example for correcting either spin-flip or phase errors). Errors other than the targeted type cannot be corrected simultaneously in the present version of the scheme. The performance of a scheme that corrects spin-flips ($\sigma_x$-errors), will for example be degraded by the presence of "complementary" $\sigma_z$-errors,

$$L_z^{(j)} = \sqrt{\Gamma_z}\sigma_z^{(j)},\qquad(31)$$

$$L_Z = \sqrt{\Gamma_Z}\sum_{j=1}^{N}\sigma_z^{(j)}.\qquad(32)$$

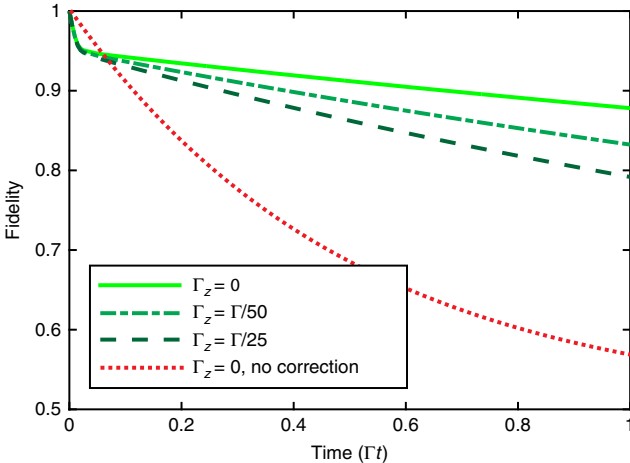

**Fig. 8** Dissipative correction of spin-flips in the presence of dephasing. The plot shows the decrease in fidelity (compare Fig. 5) for three different values of the ratio $\Gamma_z/\Gamma$, where $\Gamma_z$ ($\Gamma$) is the rate at which an individual phase-flip (individual spin-flip) occurs, assuming a sideband coupling $G = 5000\Gamma$: $\Gamma_z = 0$ (light green solid), $\Gamma_z = 1/25\Gamma$ (long green dash), and $\Gamma_z = 1/50\Gamma$ (short dark green dash). As reference, the plot also includes the decay of a single uncorrected ion under spin-flips at a rate $\Gamma$ with $\Gamma_z = 0$ (red dashed line), for which one observes much stronger decay of the fidelity

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

These errors contribute to the effective error-correcting dynamics and cause a decaying envelope for the population of the codeword $|\psi\rangle$, resulting in an additional decay of the fidelity $F_{comp}(t) = F(t)e^{-(3\Gamma_z + \Gamma)t}$. To limit the additional error to the few-percent level at $t \sim 1/\Gamma$, it is thus required that $\Gamma_k \lesssim \Gamma/50 - \Gamma/100$ depending on the kind of error process. In Fig. 8, we plot the evolution for $G/\Gamma = 5000$ and individual phase-flips with $\Gamma_z/\Gamma = 0, 1/50, 1/25$, finding that $\Gamma_z/\Gamma = 1/50$ leads to a decrease in fidelity of about ~0.05 at $t \sim 1/\Gamma$. The decrease is less pronounced for smaller $G/\Gamma$. In the absence of correction, even without dephasing ($\Gamma_z/\Gamma = 0$), one observes much stronger decay of the fidelity.

Next, we address the effect of decoherence associated with the decay of the excited degrees of freedom. More specifically, we include the decay of the excited states $|e\rangle$ and $|f\rangle$ by spontaneous emission described by the jump operators $L_{mn}^{(j)} = \sqrt{\gamma_{m,n}}|m\rangle_j\langle n|$, where $m \in \{e, f\}$ and $n \in \{0,1\}$, in the master equation. Also the motional modes are assumed to undergo decay $L_r = \sqrt{\kappa_r}r$, where $r \in \{a, b\}$. For simplicity we assume $\gamma_{m0} = \gamma_{m1} = \gamma_m/2$ (for $m \in \{e, f\}$), $\gamma_e = \gamma_f = \gamma$ and $\kappa_a = \kappa_b$. These imperfections do not change the effective couplings significantly. Analytically, they can easily be taken into account as imaginary parts in the detunings[57]: $\Delta \to \Delta - i\gamma_m/2$ and $\delta \to \delta - i\kappa_r/2$. For $\kappa_{eng} \gg \gamma_m, \kappa_r$ we can safely assume that the rates $\kappa_{eff,1-3}$ are not affected. Numerical simulations show that for the parameters used in Fig. 5, $\{G/\gamma_m, G/\kappa_r\} \sim 10^3$ or $\{\gamma_m, \kappa_r\} \sim \Gamma$, respectively, can be tolerated, as they only lead to minor errors at the percent level. Note that in the quantum jump formalism, dephasing of states $|e\rangle$ and $|f\rangle$ enter into the effective non-Hermitian Hamiltonian in a similar manner as the decay. Decay and dephasing thus only differ in their dynamics after a decay causing an error; we therefore obtain similar results for dephasing.

So far, the motional modes have been assumed to be in the ground state. Initial excitations due to imperfect cooling to mean phonon numbers $\bar{n}_r$ ($r \in \{a, b, c^{(j)}\}$), and heating processes with rates $\kappa_r^+$ constitute another source of imperfections. We study their effect in detail in Supplementary Note 3. For the auxiliary modes $c^{(j)}$, we find that neither heating nor initial excitations have a significant effect, provided that the modes are subject to a strong and continuous cooling process, such as sympathetic cooling or ancilla reset. For realistic experimental parameters $\kappa_r^+ \sim 10^{-3}G$[58,59] one obtains errors ~0.01 per mode. For the interrogation modes $a$ and $b$, initial excitations lead to an additional error $\varepsilon \approx \bar{n}_a + \bar{n}_b$. Heating of the interrogation modes has a more pronounced effect since motional excitations can couple to excitations of the system ions by the strong sideband coupling $H_{int,G}$. While such imperfections of the interrogation modes can also be countered by cooling with a rate $\kappa_r \gg \kappa_r^+$, the cooling rates need to be moderate, $\kappa_r \ll G$, to avoid decoherence during the error correction process. As detailed in Supplementary Note 3, for realistic parameters[59] the resulting errors can be at the few-percent level, comparable to the initial drop $E_0$. Still, with these parameters the presented scheme yields a significant improvement over the uncorrected case, allowing for its application in quantum metrology.

**Data availability**. The data that support the findings of this study are available from the authors upon request.

36. Morigi, G. et al. Dissipative quantum control of a spin chain. *Phys. Rev. Lett.* **115**, 200502 (2015).

37. Reiter, F., Reeb, D. & Sørensen, A. S. Scalable preparation of many-body entanglement. *Phys. Rev. Lett.* **117**, 040501 (2016).

38. Krauter, H. et al. Entanglement generated by dissipation and steady state entanglement of two macroscopic objects. *Phys. Rev. Lett.* **107**, 080503 (2011).

39. Barreiro, J. T. et al. An open-system quantum simulator with trapped ions. *Nature* **470**, 486–491 (2011).

40. Lin, Y. et al. Dissipative production of a maximally entangled steady state of two quantum bits. *Nature* **504**, 415–418 (2013).

41. Shankar, S. et al. Autonomously stabilized entanglement between two superconducting quantum bits. *Nature* **504**, 419–422 (2013).

42. Kienzler, D. et al. Quantum harmonic oscillator state synthesis by reservoir engineering. *Science* **347**, 52–56 (2015).

43. Paz, J. P. & Zurek, W. H. Continuous error correction. *Proc. R. Soc. Lond. A* **454**, 355–364 (1998).

44. Ahn, C., Doherty, A. C. & Landahl, A. J. Continuous quantum error correction via quantum feedback control. *Phys. Rev. A* **65**, 042301 (2002).

45. Sarovar, M. & Milburn, G. J. Continuous quantum error correction by cooling. *Phys. Rev. A* **72**, 012306 (2005).

46. Oreshkov, O. & Brun, T. A. Continuous quantum error correction for non-Markovian decoherence. *Phys. Rev. A* **76**, 022318 (2007).

47. Mabuchi, H. Continuous quantum error correction as classical hybrid control. *New J. Phys.* **11**, 105044 (2009).

48. Ippoliti, M., Mazza, L., Rizzi, M. & Giovannetti, V. A perturbative approach to continuous-time quantum error correction. *Phys. Rev. A* **91**, 042322 (2015).

49. Pastawski, F., Clemente, L. & Cirac, J. I. Quantum memories based on engineered dissipation. *Phys. Rev. A* **83**, 012304 (2011).

50. Fujii, K., Negoro, M., Imoto, N. & Kitagawa, M. Measurement-free topological protection using dissipative feedback. *Phys. Rev. X* **4**, 041039 (2014).

51. Kapit, E., Chalker, J. T. & Simon, S. H. Passive correction of quantum logical errors in a driven, dissipative system: a blueprint for an analog quantum code fabric. *Phys. Rev. A* **91**, 062324 (2015).

52. Kapit, E. Hardware-efficient and fully autonomous quantum error correction in superconducting circuits. *Phys. Rev. Lett.* **116**, 150501 (2016).

53. Cohen, J. & Mirrahimi, M. Dissipation-induced continuous quantum error correction for superconducting circuits. *Phys. Rev. A* **90**, 062344 (2014).

54. Freeman, C. D., Herdman, C. M. & Whaley, K. B. Engineering autonomous error correction in stabilizer codes at finite temperature. *Phys. Rev. A* **96**, 012311 (2017).

55. Shen, C. et al. Quantum channel construction with circuit quantum electrodynamics. *Phys. Rev. B* **95**, 134501 (2017).

56. Leghtas, Z. et al. Confining the state of light to a quantum manifold by engineered two-photon loss. *Science* **347**, 853–857 (2015).

57. Reiter, F. & Sørensen, A. S. Effective operator formalism for open quantum systems. *Phys. Rev. A* **85**, 032111 (2012).

58. Schindler, P. et al. A quantum information processor with trapped ions. *New J. Phys.* **15** (2013).

59. Hempel, C. *Digital Quantum Simulation, Schrödinger Cat State Spectroscopy and Setting Up a Linear Ion Trap* (PhD thesis, Leopold-Franzens-Universität Innsbruck, 2014).

60. Wunderlich, H., Wunderlich, C., Singer, K. & Schmidt-Kaler, F. Two-dimensional cluster-state preparation with linear ion traps. *Phys. Rev. A* **79**, 052324 (2009).

61. Alonso, J., Leupold, F. M., Keitch, B. C. & Home, J. P. Quantum control of the motional states of trapped ions through fast switching of trapping potentials. *New J. Phys.* **15**, 023001 (2013).

62. Plenio, M. B. & Huelga, S. F. Sensing in the presence of an observed environment. *Phys. Rev. A* **93**, 032123 (2016).

63. Wiese, U.-J. Ultracold quantum gases and lattice systems: quantum simulation of lattice gauge theories. *Ann. Phys.* **525**, 777 (2013).

64. Martinez, E. et al. Real-time dynamics of lattice gauge theories with a few-qubit quantum computer. *Nature* **534**, 516–519 (2016).

65. Vollbrecht, K. G. H., Muschik, C. A. & Cirac, J. I. Entanglement distillation by dissipation and continuous quantum repeaters. *Phys. Rev. Lett.* **107**, 120502 (2011).

66. Itano, W. H. et al. Quantum projection noise: population fluctuations in two-level systems. *Phys. Rev. A* **47**, 3554 (1993).

67. Wineland, D. J., Bollinger, J. J., Itano, W. M., Moore, F. L. & Heinzen, D. J. Spin squeezing and reduced quantum noise in spectroscopy. *Phys. Rev. A* **46**, R6797 (1992).

68. Wineland, D. J., Bollinger, J. J., Itano, W. M. & Heinzen, D. J. Squeezed atomic states and projection noise in spectroscopy. *Phys. Rev. A* **50**, 67 (1994).

## Acknowledgements

We gratefully acknowledge discussions with Philipp Schindler, Esteban Martinez, Thomas Monz, Martin van Mourik, Rainer Blatt, and Wolfgang Dür. Research was sponsored by the Army Research Laboratory and was accomplished under Cooperative Agreement Number W911NF-15-2-0060. The views and conclusions contained in this document are those of the authors and should not be interpreted as representing the official policies, either expressed or implied, of the Army Research Laboratory or the U.S. Government. This work was also supported by the European Research Council under the European Union's Seventh Framework Programme (FP/2007-2013), and the ERC Grant Agreement n. 306576. FR acknowledges support by a Feodor-Lynen fellowship from the Humboldt Foundation.

## Author contributions

All authors contributed to all aspects of this work.

## Additional information

**Competing interests:** The authors declare no competing financial interests.

