## [Peer Review File · Nature Communications]

Reviewers' comments:

Reviewer #1 (Remarks to the Author):

In the following manuscript the Authors put forward a novel proposal on how to dynamically implement an error correction protocol by controlled engineering of dissipation in an open quantum system, which then allows to counterbalance the detrimental impact of other, uncontrolled, sources of noise. In particular, they consider a system consisting of three ultra-cold ions in a trap, in order to propose a realisation of the so-called "repetition code", which is then possible thanks to cunningly designing interactions of the atoms as well as the coupling to their motional modes. They prove the proposed scheme to effectively stabilise the effective space of three physical qubits, implemented by the ions, and provide a logical qubit subspace that can be then made robust to either spin-flip or phase-flip, or correlated dephasing-type noises affecting the ions. Finally, they demonstrate how the scheme may be utilised in a quantum sensing scenario, in which it allows to correct all the dominant errors apart from the ones directly associated with the sensed-parameter encoding, which, in fact, are the ones that cannot be generally corrected without affecting the sensing resolution.

I find the presented results very interesting, with a clear potential to start a new direction in theoretical research of designing quantum error correction (QEC) protocols biased towards realistic experimental setups. In particular, the work provides a change of paradigm on how to implement QEC in ion-based systems---with, however, natural generalisation to other dissipative interacting quantum systems such as nitrogen-vacancy (NV) centres, opto-mechanical systems or superconducting qubits---which, as far as I know, has been typically approached by studying implementations of quantum gates and measurements accompanied by feedback rather than dissipative dynamics. On the other hand, the Authors provide a very thorough analysis of the proposed scheme, importantly including its robustness to other unavoidable imperfections. As a result, I think that the presented ideas will not only spark theoretical work on the subject, but also provide a platform that may be directly used by the experimentalists. That is why, I am tempted to recommend the article for publication in the Nature Communications journal. However, I find that the clarity and presentation of the results must still be greatly improved with the manuscript requiring a major revision. In what follows, I list one by one, in the order of their appearance in text, the issues that I urge the Authors to address for the paper to be later acceptable for publication.

1) I think that the use of word "autonomous" throughout the manuscript is inappropriate. In order to be sure, I have gone through the cited literature on "dissipative QEC" and have not come across such a nomenclature, thus, I presume that it has been introduced by the Authors. In my opinion, word "autonomous" is normally reserved in other fields, e.g. "autonomous machines/learning/robots", to describe systems that are capable to adapt while gaining information about their surroundings during the evolution. Hence, wording "autonomous" QEC for me naturally suggests that there is some subpart of a given quantum system that can "adaptively" correct for the

errors by adjusting and "autonomously" learning what error mechanisms affect the whole system. This is clearly not the case here, so I urge the Authors to avoid such phrasing (including the figures). In fact, I find it very important, as one of the key features of the presented work is the paradigm that: one designs/engineers (but fixes!) dissipative Lindbladians of a large quantum system, so that there is a subspace (manifold) that is then decoherence-free with respect to other forms of dissipation. In particular, one first knows what sort of noises to fight against and then "pre-designs" the protocol. On the other hand, I find the two natures of the put-forward QEC to be crucial that should be clearly emphasised in the introduction and the title: its "dynamical/continuous" (rather than static gate-based) and "dissipative" (implementation with Lindbladians rather than unitaries as in, e.g., dynamical decoupling) characters. Potential examples of titles that quickly strike my mind are: "Dynamical quantum error correction by engineering the dissipation of trapped ions with application to sensing" or "Dissipative quantum error correction with trapped ions and its application to quantum sensing".

2) The introduction adequately refers to previous works when introducing the idea of QEC in sensing and comparing it against the dynamical control methods. However, I think that the Authors need to be more explicit in referring to very recent implementations of QEC for quantum metrology with NV-centres. It is so, as their work will be naturally compared against these previous results. In particular, I think that they should stress that the NVs have been shown to be very promising as a platform to implement QEC within the "standard" (non-dissipative) EC paradigm [see also doi:10.1038/nature12919, doi:10.1038/NNANO.2014.2]. Moreover, such approach have recently been successfully used to implement QEC in the context of quantum sensing with NVs, see Ref. [15]. Yet, the "dissipative" EC paradigm that the Authors elaborate on has not been up to now explored and, in particular, it seems very promising in case of ultra-cold ions, as shown explicitly within the presented work.

3) Regarding the theoretical work on quantum metrology with EC, let me share my general feeling that, after the doi:10.1103/PhysRevLett.113.250801 publication (which Authors may consider also citing), the community considered all the results of Refs. [10-12,18, (also doi:10.1103/PhysRevX.5.031010)] to be very unique and restricted, as they all have dealt with a special type of errors, i.e., noise that is directed perpendicularly to the sensed-parameter encoding. Nevertheless, this scenario has been successfully implemented in Ref. [15] with NVs to correct the "perpendicular noise". Ref. [14] (despite being limited to qubits) is quite important as it has showed that the protocol utilised in Ref. [15] can completely restore the so-called Heisenberg scaling (quadratic scaling of mean-squared error) in time also for other directed noise-types with the only exception---when the noise direction coincides exactly! with the parameter encoding. In that sense, Ref. [14] was first to show that there is room for, also "dissipative", QEC to be effective not only when dealing with perpendicular noise.

4) In the last sentence of the second paragraph on page 2: "this allows one to stabilize a qubit continuously against single-qubit spin flips or phase flips or against correlated noise", please change

it to "this allows one to stabilize a qubit continuously against a given type of error: either single-qubit spin flips or phase flips, or against correlated noise".

5) Between Eq. (2) and Eq. (3) please stress that the L_k jump operators can be multi-qubit. For instance, consider rewriting "Local spin flips..." to something a'la "In the special case when L_k describe local spin-flip noise affecting independently each of the constituent qubits...".

6) Please number all the Equations including ones in the Appendices.

7) I urge the Authors to change their notation of operator subscripts and superscripts. The one they have adopted is very confusing, so that the reader quickly loses track whether the index: labels the qubit number, refers to one of the Pauli operators, or is just numbering the operator within a set. For example, considering Eqs. (1-3), one has $\{\sigma_j^x, L_k \text{ and } L_{\{x_j\}}\}$ with $\{k,j\}$ crucially referring to different things. Note also that it is sometime common to write σ_j ($j=1,2,3$) as σ_j ($j=x,y,z$) for the 3 Pauli operators, and the discussed protocol unfortunately employs exactly 3 qubits, what makes things even worse.

I propose the Authors to adopt a standard (to me) notation such that Eq.1 reads

$$\langle \psi^{(j)} | \sigma_x^{(j)} | \psi \rangle, \quad j=1,2,3,$$

while Eq.3 reads

$$L_{x^{(j)}} = \sqrt{\Gamma} \sigma_x^{(j)}, \quad j=1,2,3.$$

Then, the two-body stabiliser operators S can be represented as

$$S^{(12)} = \sigma_z^{(1)} \sigma_z^{(2)}.$$

I am aware that this may cause trouble later when the Authors introduce extra "subscripts" such as "corr" in Eq. (4), but I insist and I am sure they can find a solution, e.g., in Eq. (4):

$$L_{x, \text{corr}}^{(2)}$$

[N.B. Considering changing "corr" shorthand to "e.c." throughout the manuscript, as may be naturally mistaken for "correlated" (as in correlated noise) which creates even more confusion.]

8) Consistently, above Eq. (4) "autonomous" should be changed to "dissipative".

9) Similarly, $\sigma_{\text{pm}}^{(j)}$ in Eq. (7).

10) In the caption of Fig. 3, the wording "three system ions and two oscillator modes a and b" is dubious and should be improved to, e.g., "three system ions, each of which is coupled to separate two oscillator modes a and b", as one needs in principle two auxiliary oscillator modes (or an extra ancillary ion) per each system ion. However, as in the case of correlated noise (as discussed end of the Section III A) one requires two auxiliary modes for the whole system, it may be better to just describe the case of a single ion in the figure caption.

11) Above Eq. (9), please consider changing "Implementations..." to, e.g., "However, the analysis that follows can be straightforwardly adopted to the case of ancilla ions by replacing..." so that it is clear that it refers to text that follows. Consistently, please also change there the jump operator notation there to $L_{-}^{(j)} = \sqrt{\kappa} \sigma_{-}^{(j)}$.

12) In all the Hamiltonians in the manuscript (in particular Eqs. (one before 8,8-12), I insist to include in their notation the coupling constant besides their description. For instance, I suggest strongly the Authors to use the labelling $H_{g}^{\{\mathrm{anc}\}}$ or $H_{\{\mathrm{anc}\}}^{g}$ for the coupling to ancillary modes in Eq. (9) [and similarly for all the other Hamiltonians]. Note that this makes a big difference to the reader, as he/she can have a look at Figs. 3-5 and directly see from each coupling label (δ, g, Δ) which Hamiltonian describes the corresponding interaction.

13) Below Eq. (12), if I am correct, it would be already worth mentioning that the coherence that is preserved in that way (between $|e\rangle$ and $|f\rangle$) can be designed to be the one required to precisely sense a parameter in the quantum metrology setting.

14) At the beginning of the "Working Mechanism" section, the Authors should promote the first sentence to a short paragraph. In particular, within it, they should summarize what follows in the section, i.e., firstly the description of how to implement the dissipative QEC of Eq.(7) with the aforementioned couplings. Yet, in its second part it should be already explicitly stated that the couplings will also (unfortunately) introduce other type of "undesired" dynamics, which, however, can be dealt with (please, explicitly describe, e.g., by slowing their down, optimisation of coupling parameters for double-error events etc.) by further interaction engineering described in the second part of the section.

15) I think that after the statement "We remark that the correction of several types of errors can also be realized sequentially rather than simultaneously..." on page 5, it would be worthwhile, if I am correct, to comment that such sequential procedure would be inferior, in the sense that other "external imperfections" (such as the ones considered in Section IV.A) would have a stronger impact due to the error correcting sequences taking longer times. Furthermore, please correct me if I am wrong, I have the intuition that when effectively "Trotterising" and correcting for errors on qubit 1,

over period T_1 , one needs to then assume that no errors will occur afterwards during T_2 and T_3 . In that sense, the dissipative QEC applied on the first qubit during T_1 must be sufficient for the whole duration $T_1+T_2+T_3$.

16) When introducing the effective correcting jump-operator for phase flips at the end of Section III A on page 6, it would be nice (now, given the improved notation) to explicitly refer to jump-operators in Eq. (4-6).

17) Moreover, I think that the last sentence of Section III A, which discusses the jump-operator design for the correction of collective phase-flips, should be promoted to a short paragraph with more explicit explanation.

18) On page 7 (second paragraph), reading the sentence: "it features a fast initial drop to a fidelity ... which is characteristic for time-continuous quantum error-correcting schemes.", I was unsure whether this is a general feature of continuous EC protocols---in such a case please provide a reference---or a feature discovered by the Authors in App. B2---in such a case it would be nice to see some argument that it is not a consequence of a particular approach that the Authors take, in particular, the "rate equation model" they adopt.

19) In Fig. 7, please use the same naming: either "spin-flip" or "bit-flip", throughout. In the legend of the figure, please write " $\gamma_z=0$, no correction", while in the caption consider changing the last sentence (and maybe elaborating in text), in order to stress that the red curve is plotted to show that in the absence of correction, even without dephasing, one observes much stronger decays of fidelity.

20) In Section V, within the paragraph on page 8 starting "In the presence of noise, the precision of the measurement changes significantly [5, 18]" I think the Authors should also refer to canonical references on noisy metrology, i.e., doi:10.1038/nphys1958 and doi:10.1038/ncomms2067.

21) There are two issues regarding the section "Application to quantum metrology" that I would like the Authors to resolve/discuss for me to recommend their work for publication:

a. They consider the Ramsey-type protocol, but another solution is to measure the "parity" operator which in absence of noise also yields the Heisenberg scaling (linear in time) of precision for GHZ states, while in the case of transverse noise it has been shown to also attain the optimal scaling discussed in App. C2 [doi:10.1103/PhysRevX.5.031010]. Could the parity operator be implemented

with ion traps especially when considering many logical qubits and the segmented-traps implementation discussed in the last paragraph?

b. The authors have considered to correct for spin(bit)-flip errors generated by σ_x in the case when the parameter is encoded by σ_z . Following Ref. [14], which proved that any rank-one Pauli noise is correctable---not just the transversal case---I expect that if the Authors tilted the direction of H from σ_z to a general $\sigma_{\mathbf{n}} = n_x \sigma_x + n_y \sigma_y + n_z \sigma_z$, where $\mathbf{n} = \{n_x, n_y, n_z\}$ is any unit vector (but not $\{0,0,1\}$ for exactly parallel dephasing), then they would still observe linear scaling of precision in time and large improvement due to correction. However, they would need to prepare their 3 qubits in the GHZ state with respect to the tilted \mathbf{n} basis, and also perform the Ramsey sequence in this basis. My guess is that (following Ref. [14]) the precision will be rescaled by a constant factor, i.e., $\sim \sin(\text{angle between } \mathbf{n} \text{ and } x)$, what will not affect the scaling in t in Fig. 8 at short times. I think that the Authors should verify this, as it will greatly improve their result---proving that the proposed dissipative QEC is not restricted to just perpendicular qubit noise.

22) The second paragraph of Section V refers to App. C2 in two places. It seems to me that it should refer to App. C and Fig.11 therein, in which the impact of collective dephasing is also discussed.

23) One of the outlooks that the Authors may consider in the concluding section is whether one could combine the paradigm of dissipative QEC with the technique of monitoring environmental degrees of freedom, as proposed in doi:10.1103/PhysRevA.93.032123. As the dissipative QEC will not be able to correct, e.g., for losses, but only rank-one Pauli noise-types (and collective dephasings) [see Ref. [14]], maybe a hybrid scheme that also benefits from observing some environmental operators would constitute the ultimate goal for QEC.

24) Please amend throughout the Appendices the sub-/super-script notation of operators according to the main text and point 7) above. Note that in Appendix A1 it becomes even more confusing to use the labelling L_{corr} , as this naturally suggests to refer to the "correlated spin flips" there discussed.

25) Typos "correlate" -> "correlated" below (A1) and "an"->"a" above (A6).

26) In the Equation (B16) which value of parameter "r" did the Authors use? Is it the 2.5 suggested below Eq. (B8)? Please clarify.

27) In the discussion of Appendix C2, and the case of transverse noise with no error-correction, the Authors should note that the numerical constant c (defined in the left column of page 17) depends also on the value of the estimated parameter ω [arxiv version of Ref [18] and doi:10.1103/PhysRevX.5.031010]. In particular, it vanishes at $\omega=0$ for which the exact Heisenberg scaling can be achieved [doi:10.1103/PhysRevX.5.031010].

Reviewer #2 (Remarks to the Author):

The manuscript is a thorough study into how bit or phase flip errors on a logical qubit can be corrected via a dissipative scheme. The process constitutes a dissipative version of the three qubit repetition code. The authors also discuss how such correction can be applied to quantum sensing.

The manuscript considers a self-correction process, but only one of limited scope. Only three qubits are used in the code, and only one type of error is corrected. This means that the best correction operator can be determined by looking at only two syndromes. It is such properties that allow their scheme to be implemented.

As such, I don't see that the work will be of great interest to those working of quantum error correction or self-correction for large scale devices. Instead it will interest those interested in error mitigation techniques in smaller scale set-ups, such as the quantum sensing discussed at the end of the manuscript.

I believe the manuscript is technically correct and of high quality. It is certainly suitable for publication in a good journal. However, I must defer to the opinion of someone within the quantum sensing community to determine whether it is of sufficient impact for Nature Communications.

Reviewer #3 (Remarks to the Author):

The manuscript by Reiter et al. "Autonomous Quantum Error Correction and Application to Quantum Sensing with Trapped Ions" is very interesting, novel and exciting.

Trapped ions is a mature technology both for quantum computing/quantum information applications and for precision measurements, including the frequency standards. The proposed

dissipative method for correcting bit-flip errors (and its extensions to correct phase-flip errors and correlated errors) is an important step towards reliable quantum sensing, as well as possible avenue towards scalable quantum computing.

One drawback that I see in the proposed scheme as an experimentalist is the necessity to cool down to the ground state. While nearly-achievable, ground state cooling is never perfect, and heating is inevitable during operation time (especially if the interrogation time is long). Since vibrational modes play such an important role in the scheme, it would be informative to see the effects of imperfect cooling and subsequent heating mentioned in the paper. Would continuous sympathetic cooling be an useful option.

Overall, the paper is extremely well written and referenced, and I recommend its publication provided that my above question is addressed.

Response to Reviewer #1:

We thank Reviewer #1 for the very careful reading of our manuscript and for his/her detailed comments on our work.

The Referee writes:

In the following manuscript the Authors put forward a novel proposal on how to dynamically implement an error correction protocol by controlled engineering of dissipation in an open quantum system, which then allows to counterbalance the detrimental impact of other, uncontrolled, sources of noise. In particular, they consider a system consisting of three ultra-cold ions in a trap, in order to propose a realisation of the so-called "repetition code", which is then possible thanks to cunningly designing interactions of the atoms as well as the coupling to their motional modes. They prove the proposed scheme to effectively stabilise the effective space of three physical qubits, implemented by the ions, and provide a logical qubit subspace that can be then made robust to either spin-flip or phase-flip, or correlated dephasing-type noises affecting the ions. Finally, they demonstrate how the scheme may be utilised in a quantum sensing scenario, in which it allows to correct all the dominant errors apart from the ones directly associated with the sensed-parameter encoding, which, in fact, are the ones that cannot be generally corrected without affecting the sensing resolution.

I find the presented results very interesting, with a clear potential to start a new direction in theoretical research of designing quantum error correction (QEC) protocols biased towards realistic experimental setups. In particular, the work provides a change of paradigm on how to implement QEC in ion-based systems---with, however, natural generalisation to other dissipative interacting quantum systems such as nitrogen-vacancy (NV) centres, opto-mechanical systems or superconducting qubits---which, as far as I know, has been typically approached by studying implementations of quantum gates and measurements accompanied by feedback rather than dissipative dynamics. On the other hand, the Authors provide a very thorough analysis of the proposed scheme, importantly including its robustness to other unavoidable imperfections. As a result, I think that the presented ideas will not only spark theoretical work on the subject, but also provide a platform that may be directly used by the experimentalists. That is why, I am tempted to recommend the article for publication in the Nature Communications journal. However, I find that the clarity and presentation of the results must still be greatly improved with the manuscript requiring a major revision. In what follows, I list one by one, in the order of their appearance in text, the issues that I urge the Authors to address for the paper to be later acceptable for publication.

Response to the Referee:

We are grateful to the Referee for this encouraging judgement of our work, as well as for the helpful feedback below, which has helped us to improve our manuscript.

The Referee writes:

1) I think that the use of word "autonomous" throughout the manuscript is inappropriate. In order to be sure, I have gone through the cited literature on "dissipative QEC" and have not come across such a nomenclature, thus, I presume that it has been introduced by the Authors. In my opinion, word "autonomous" is normally reserved in other fields, e.g. "autonomous machines/learning/robots", to describe systems that are capable to adapt while gaining information about their surroundings

during the evolution. Hence, wording "autonomous" QEC for me naturally suggests that there is some subpart of a given quantum system that can "adaptively" correct for the errors by adjusting and "autonomously" learning what error mechanisms affect the whole system. This is clearly not the case here, so I urge the Authors to avoid such phrasing (including the figures). In fact, I find it very important, as one of the key features of the presented work is the paradigm that: one designs/engineers (but fixes!) dissipative Lindbladians of a large quantum system, so that there is a subspace (manifold) that is then decoherence-free with respect to other forms of dissipation. In particular, one first knows what sort of noises to fight against and then "pre-designs" the protocol. On the other hand, I find the two natures of the put-forward QEC to be crucial that should be clearly emphasised in the introduction and the title: its "dynamical/continuous" (rather than static gate-based) and "dissipative" (implementation with Lindbladians rather than unitaries as in, e.g., dynamical decoupling) characters. Potential examples of titles that quickly strike my mind are: "Dynamical quantum error correction by engineering the dissipation of trapped ions with application to sensing" or "Dissipative quantum error correction with trapped ions and its application to quantum sensing".

Response to the Referee:

We agree that the use of the term "autonomous" is not fully appropriate, for the reasons given by the Referee. Throughout our revised manuscript, including the title, the abstract, the introduction, and the figures, we have replaced the term "autonomous" by the suggested words "dissipative", "dynamical" and "continuous", and emphasized the importance of the latter.

The Referee writes:

2) The introduction adequately refers to previous works when introducing the idea of QEC in sensing and comparing it against the dynamical control methods. However, I think that the Authors need to be more explicit in referring to very recent implementations of QEC for quantum metrology with NV-centres. It is so, as their work will be naturally compared against these previous results. In particular, I think that they should stress that the NVs have been shown to be very promising as a platform to implement QEC within the "standard" (non-dissipative) EC paradigm [see also doi:10.1038/nature12919, doi:10.1038/NNANO.2014.2]. Moreover, such approach have recently been successfully used to implement QEC in the context of quantum sensing with NVs, see Ref. [15]. Yet, the "dissipative" EC paradigm that the Authors elaborate on has not been up to now explored and, in particular, it seems very promising in case of ultra-cold ions, as shown explicitly within the presented work.

Response to the Referee:

We are grateful to the Referee for pointing out these important works on quantum-error correction-enhanced sensing with NV centers, as well as for the positive remarks on the context of our work. We agree that this previous work is highly relevant to our approach. In the revised version of our manuscript, we highlight the achievements with NV centers.

The Referee writes:

3) Regarding the theoretical work on quantum metrology with EC, let me share my general feeling that, after the doi:10.1103/PhysRevLett.113.250801 publication (which Authors may consider also citing), the community considered all the results of Refs. [10-12,18, (also

doi:10.1103/PhysRevX.5.031010]) to be very unique and restricted, as they all have dealt with a special type of errors, i.e., noise that is directed perpendicularly to the sensed-parameter encoding. Nevertheless, this scenario has been successfully implemented in Ref. [15] with NVs to correct the "perpendicular noise". Ref. [14] (despite being limited to qubits) is quite important as it has showed that the protocol utilised in Ref. [15] can completely restore the so-called Heisenberg scaling (quadratic scaling of mean-squared error) in time also for other directed noise-types with the only exception---when the noise direction coincides exactly! with the parameter encoding. In that sense, Ref. [14] was first to show that there is room for, also "dissipative", QEC to be effective not only when dealing with perpendicular noise.

Response to the Referee:

We thank the Referee for pointing out these important works and aspects. In the revised version, we added the suggested references and emphasized these connections in the section "Application to quantum metrology".

The Referee writes:

4) In the last sentence of the second paragraph on page 2: "this allows one to stabilize a qubit continuously against single-qubit spin flips or phase flips or against correlated noise", please change it to "this allows one to stabilize a qubit continuously against a given type of error: either single-qubit spin flips or phase flips, or against correlated noise".

5) Between Eq. (2) and Eq. (3) please stress that the L_k jump operators can be multi-qubit. For instance, consider rewriting "Local spin flips..." to something a'la "In the special case when L_k describe local spin-flip noise affecting independently each of the constituent qubits...".

6) Please number all the Equations including ones in the Appendices.

7) I urge the Authors to change their notation of operator subscripts and superscripts. The one they have adopted is very confusing, so that the reader quickly loses track whether the index: labels the qubit number, refers to one of the Pauli operators, or is just numbering the operator within a set. For example, considering Eqs. (1-3), one has $\{\sigma_j^x, L_k \text{ and } L_{\{x_j\}}\}$ with $\{k,j\}$ crucially referring to different things. Note also that it is sometime common to write σ_j ($j=1,2,3$) as σ_j ($j=x,y,z$) for the 3 Pauli operators, and the discussed protocol unfortunately employs exactly 3 qubits, what makes things even worse.

I propose the Authors to adopt a standard (to me) notation such that Eq.1 reads $|\left| \psi^{(j)} \right\rangle \rangle = |\sigma_{\{x\}}^{(j)} \rangle \rangle$, $\quad j=1,2,3$, while Eq.3 reads

$$L_{\{x\}}^{(j)} = \sqrt{\Gamma} \sigma_{\{x\}}^{(j)}, \quad j=1,2,3.$$

Then, the two-body stabiliser operators S can be represented as

$$S^{(12)} = \sigma_z^{(1)} \sigma_z^{(2)}.$$

I am aware that this may cause trouble later when the Authors introduce extra "subscripts" such as "corr" in Eq. (4), but I insist and I am sure they can find a solution, e.g., in Eq. (4):

$$L_{\{x, \text{corr}\}}^{(2)}$$

[N.B. Considering changing "corr" shorthand to "e.c." throughout the manuscript, as may be naturally mistaken for "correlated" (as in correlated noise) which creates even more confusion.]

8) Consistently, above Eq. (4) "autonomous" should be changed to "dissipative".

9) Similarly, $\sigma_{\rho m}^{(j)}$ in Eq. (7).

Response to the Referee:

We have implemented the suggested changes and thereby improved the clarity of our manuscript.

The Referee writes:

10) In the caption of Fig. 3, the wording "three system ions and two oscillator modes a and b" is dubious and should be improved to, e.g., "three system ions, each of which is coupled to separate two oscillator modes a and b", as one needs in principle two auxiliary oscillator modes (or an extra ancillary ion) per each system ion. However, as in the case of correlated noise (as discussed end of the Section III A) one requires two auxiliary modes for the whole system, it may be better to just describe the case of a single ion in the figure caption.

Response to the Referee:

We thank the Referee for this suggestion, which has helped us to better highlight the two types of motional modes required for our scheme:

The modes a and b are in fact two shared modes, to which all three system ions couple collectively. In the revised manuscript, these are now referred to as "interrogation modes" to distinguish them from the $c^{(j)}$ -modes, to which the ions are coupled separately (when correcting for individual spin flips). Following the suggestion of the Referee, the $c^{(j)}$ -modes are now consistently termed "auxiliary modes". Beside the caption of Fig.3, we have edited the text in the Setup section to make the discussion more instructive, and also renamed H_{osc} to $H_{\text{int,G}}$ (interrogation) and H_{anc} to $H_{\text{aux,g}}$ (auxiliary) for clarity.

The Referee writes:

11) Above Eq. (9), please consider changing "Implementations..." to, e.g., "However, the analysis that follows can be straightforwardly adopted to the case of ancilla ions by replacing..." so that it is clear the it refers to text that follows. Consistently, please also change there the jump operator notation there to $L_{-}^{(j)} = \sqrt{\kappa} \sigma_{-}^{(j)}$.

Response to the Referee:

In the revised manuscript, the possibility to replace the auxiliary modes by ancilla qubits is now better explained and the notation is changed using the suggestions of the Referee.

The Referee writes:

12) In all the Hamiltonians in the manuscript (in particular Eqs. (one before 8,8-12), I insist to include in their notation the coupling constant besides their description. For instance, I suggest strongly the Authors to use the labelling $H_{\text{g}}^{\text{anc}}$ or $H_{\text{anc}}^{\text{g}}$ for the coupling to ancillary modes in Eq. (9) [and similarly for all the other Hamiltonians]. Note that this makes a big difference to the reader, as he/she can have a look at Figs. 3-5 and directly see from each coupling label (δ, g, Δ) which Hamiltonian describes the corresponding interaction.

Response to the Referee:

We appreciate this helpful suggestion. We have implemented it in the revised manuscript.

The Referee writes:

13) Below Eq. (12), if I am correct, it would be already worth mentioning that the coherence that is preserved in that way (between $|e\rangle$ and $|f\rangle$) can be designed to be the one required to precisely sense a parameter in the quantum metrology setting.

Response to the Referee:

We have added a comment to the revised manuscript.

The Referee writes:

14) At the beginning of the "Working Mechanism" section, the Authors should promote the first sentence to a short paragraph. In particular, within it, they should summarize what follows in the section, i.e., firstly the description of how to implement the dissipative QEC of Eq.(7) with the aforementioned couplings. Yet, in its second part it should be already explicitly stated that the couplings will also (unfortunately) introduce other type of "undesired" dynamics, which, however, can be dealt with (please, explicitly describe, e.g., by slowing their down, optimisation of coupling parameters for double-error events etc.) by further interaction engineering described in the second part of the section.

Response to the Referee:

We agree that an initial paragraph makes the "Working mechanism" section more accessible and leads also to a better understanding of the physics behind the optimization. In the revised manuscript, we have added the introductory paragraph, summarizing the engineering of the dissipation and pointing out the additional error processes and how to counter them. In the second part of the section, we have added a discussion of the physical implications of the optimization of the protocol (which is carried out quantitatively in later sections).

The Referee writes:

15) I think that after the statement "We remark that the correction of several types of errors can also be realized sequentially rather than simultaneously..." on page 5, it would be worthwhile, if I am correct, to comment that such sequential procedure would be inferior, in the sense that other "external imperfections" (such as the ones considered in Section IV.A) would have a stronger impact due to the error correcting sequences taking longer times. Furthermore, please correct me if I am wrong, I have the intuition that when effectively "Trotterising" and correcting for errors on qubit 1, over period T_1 , one needs to then assume that no errors will occur afterwards during T_2 and T_3 . In that sense, the dissipative QEC applied on the first qubit during T_1 must be sufficient for the whole duration $T_1+T_2+T_3$.

Response to the Referee:

We agree with the Referee on these consequences of a sequential correction. We discuss them in the revised manuscript.

The Referee writes:

16) When introducing the effective correcting jump-operator for phase flips at the end of Section III A on page 6, it would be nice (now, given the improved notation) to explicitly refer to jump-operators in Eq. (4-6).

Response to the Referee:

We appreciate this comment and have implemented it.

The Referee writes:

17) Moreover, I think that the last sentence of Section III A, which discusses the jump-operator design for the correction of collective phase-flips, should be promoted to a short paragraph with more explicit explanation.

Response to the Referee:

We have extended the discussion of collective phase-flips to a paragraph.

The Referee writes:

18) On page 7 (second paragraph), reading the sentence: "it features a fast initial drop to a fidelity ... which is characteristic for time-continuous quantum error-correcting schemes.", I was unsure whether this is a general feature of continuous EC protocols---in such a case please provide a reference---or a feature discovered by the Authors in App. B2---in such a case it would be nice to see some argument that it is not a consequence of a particular approach that the Authors take, in particular, the "rate equation model" they adopt.

Response to the Referee:

We are grateful for this question which helps us to improve the discussion in the text. As we point out in the revised manuscript, the initial drop is indeed a generic feature of continuous quantum error correction approaches, e.g., Ref. [46] (Ref. [47] in the previous version). To make this clear, we have separated the discussion of the physical effects from that of the rate model.

The Referee writes:

19) In Fig. 7, please use the same naming: either "spin-flip" or "bit-flip", throughout. In the legend of the figure, please write " $\gamma_z=0$, no correction", while in the caption consider changing the last sentence (and maybe elaborating in text), in order to stress that the red curve is plotted to show that in the absence of correction, even without dephasing, one observes much stronger decays of fidelity.

Response to the Referee:

We have unified the notation to "spin-flip" throughout and implemented the suggestion in the legend of Fig. 7 and in the text.

The Referee writes:

20) In Section V, within the paragraph on page 8 starting "In the presence of noise, the precision of the measurement changes significantly [5, 18]" I think the Authors should also refer to canonical references on noisy metrology, i.e., doi:10.1038/nphys1958 and doi:10.1038/ncomms2067.

Response to the Referee:

We have added the suggested references to the manuscript.

The Referee writes:

21) There are two issues regarding the section "Application to quantum metrology" that I would like the Authors to resolve/discuss for me to recommend their work for publication:

a. They consider the Ramsey-type protocol, but another solution is to measure the "parity" operator which in absence of noise also yields the Heisenberg scaling (linear in time) of precision for GHZ states, while in the case of transverse noise it has been shown to also attain the optimal scaling discussed in App. C2 [doi:10.1103/PhysRevX.5.031010]. Could the parity operator be implemented with ion traps especially when considering many logical qubits and the segmented-traps implementation discussed in the last paragraph?

Response to the Referee:

Yes, there are at least two methods to implement the "parity" operator as discussed in [doi:10.1103/PhysRevX.5.031010].

Method 1:

The parity operator can be straightforwardly and efficiently be implemented in trapped ions, both in the case of a single logical qubit and in the case of many logical qubits in a segmented trap.

The procedure for implementing the parity operator for a single logical qubit

$$A_x = \sigma_x^{(1)} \sigma_x^{(2)} \sigma_x^{(3)}$$

is described in detail in [New Journal of Physics 13, 085007 (2011)] in Sec. 3. The scheme has been experimentally demonstrated for four system ions in Innsbruck as described in [Nature 470, 486 (2011)]. The procedure requires an additional ancilla ion. By applying (1) a standard Mølmer-Sørensen interaction to all ions, (ii) a single qubit rotation on the ancilla and (iii) another Mølmer-Sørensen interaction on all ions, the Hamiltonian $H = \omega A_x \sigma_y^{(Ancilla)}$ can be implemented which allows one to measure the parity operator A_x (acting on the system ions) by a measurement on the ancilla ion.

Importantly, the energy scale ω is essentially independent on the number of system ions (see [New Journal of Physics 13, 085007 (2011)], Sec. 3.3). This allows for a generalization of the approach to N logical qubits. For measuring the parity operator, the potential barriers of the segmented trap have to be ramped down such that all ions share a common motional mode that can be used to perform the required Mølmer-Sørensen gates.

Method 2:

An even simpler implementation can be realized as follows. In the case of a single logical qubit, the global (3-qubit)- $\pi/2$ pulse at the end of the Ramsey sequence is replaced by local $\pi/2$ pulses on each of the three physical qubits. After this operation, the parity of the resulting spin state is measured by performing three local measurements in the 0/1 basis on the individual spins. The parity P of the spin state is then given by $P = (\text{number of the spins in state } 1) \bmod 2$. This method yields Heisenberg scaling in the ideal case and is easy to realize experimentally, since single qubit operations and local measurements can be performed with very high fidelity in trapped ions. In the case of N logical qubits, the global (N -qubit)- $\pi/2$ pulse at the end of the Ramsey sequence should be replaced by quasi-local $\pi/2$ pulses that act on each logical qubit separately. Since each logical qubit consists of three physical qubits, this can be routinely done by applying the same Mølmer-Sørensen-type gate that is used for the generation of the 3-qubit GHZ states.

We included the possibility to use method 2 (local $\pi/2$ pulses and local qubit measurements) in the paper and thank the referee for his/her suggestion.

The Referee writes:

b. The authors have considered to correct for spin(bit)-flip errors generated by σ_x in the case when the parameter is encoded by σ_z . Following Ref. [14], which proved that any rank-one Pauli noise is correctable---not just the transversal case---I expect that if the Authors tilted the direction of H from σ_z to a general $\sigma_{\mathbf{n}} = n_x \sigma_x + n_y \sigma_y + n_z \sigma_z$, where $\mathbf{n} = \{n_x, n_y, n_z\}$ is any unit vector (but not $\{0,0,1\}$ for exactly parallel dephasing), then they would still observe linear scaling of precision in time and large improvement due to correction. However, they would need to prepare their 3 qubits in the GHZ state with respect to the tilted \mathbf{n} basis, and also perform the Ramsey sequence in this basis. My guess is that (following Ref. [14]) the precision will be rescaled by a constant factor, i.e., $\sim \sin(\text{angle between } \mathbf{n} \text{ and } x)$, what will not affect the scaling in t in Fig. 8 at short times. I think that the Authors should verify this, as it will greatly improve their result---proving that the proposed dissipative QEC is not restricted to just perpendicular qubit noise.

Response to the Referee:

We thank the reviewer for pointing this out. Before going into the details, we would like to mention that our scheme can correct Pauli noise in any direction if the level scheme shown in Fig. 3 is adjusted accordingly. Since this can result in a complicated scheme, we consider in the following the correction of σ_x noise in the presence of a signal that is not necessarily oriented in the z -direction.

While the argument in Ref. [14] (Ref. [24] in the revised version) is unfortunately not directly applicable to our case, we can get a similar result (i.e. a rescaling of the precision by a constant factor that is determined by the angle between \mathbf{n} and x), as anticipated by the referee [personal communication with Wolfgang Dür].

More specifically, Ref [14] assumes a fast and complete error correction scheme that is able to perfectly correct Pauli errors of a given type (for example σ_x). To this end [14] assumes that each sensing qubit is supplemented with a protected ancilla qubit which allows for a perfect correction of σ_x errors. In contrast, our three qubit code only provides an error reduction since two and three qubit spin flips cannot be corrected. Moreover experimental imperfections render the code non-perfect.

In contrast to the setup where each sensing qubit is accompanied by a perfect ancilla qubit, our three qubit code also does not allow us to prepare the GHZ state in a tilted bases while still correcting σ_x errors by a direct application of our scheme (we need to use the logical basis states $|000\rangle$ and $|111\rangle$).

However, following the intuition of the reviewer, we can make the following argument:

As suggested in the report, we consider the signal Hamiltonian

$$H = \omega * \sigma_{\mathbf{n}}$$

with

$$\sigma_{\mathbf{n}} = n_x * \sigma_x + n_y * \sigma_y + n_z * \sigma_z.$$

We consider a situation in which the GHZ state $|000\rangle + |111\rangle$ is prepared and our scheme is used to correct for spin flips.

In addition, we assume that fast unitary rotations are performed on each of the individual qubits. More specifically, the rotations $U = \exp[i(\pi/2)\sigma_z]$ and $U^\dagger = \exp[-i(\pi/2)\sigma_z]$ are applied in alternation. This leaves the z-component of the signal invariant but leads to a sign-change in the x and y components:

$$U.\sigma_x.U^\dagger = -\sigma_x,$$

$$U.\sigma_y.U^\dagger = -\sigma_y,$$

If the rotations are applied sufficiently fast, the x and the y component of the signal Hamiltonian cancel out and only the z-projection of the signal Hamiltonian plays a role. Since our error correction processes remain unaffected by the applied unitary rotations, we obtain a result which resembles in spirit the finding in [14], showing that the proposed dissipative QEC is not restricted to just perpendicular qubit noise.

We added a corresponding statement in the section "Application to Quantum Metrology" and included an explanation (see reference/footnote [57] in the revised manuscript).

The Referee writes:

22) The second paragraph of Section V refers to App. C2 in two places. It seems to me that it should refer to App. C and Fig.11 therein, in which the impact of collective dephasing is also discussed.

Response to the Referee:

We thank the referee for pointing out this mistake and have corrected it in the revised manuscript.

The Referee writes:

23) One of the outlooks that the Authors may consider in the concluding section is whether one could combine the paradigm of dissipative QEC with the technique of monitoring environmental degrees of freedom, as proposed in doi:10.1103/PhysRevA.93.032123. As the dissipative QEC will not be able to correct, e.g., for losses, but only rank-one Pauli noise-types (and collective dephasings)

[see Ref. [14]], maybe a hybrid scheme that also benefits from observing some environmental operators would constitute the ultimate goal for QEC.

Response to the Referee:

We have added these ideas to the conclusions of our revised manuscript.

The Referee writes:

24) Please amend throughout the Appendices the sub-/super-script notation of operators according to the main text and point 7) above. Note that in Appendix A1 it becomes even more confusing to use the labelling L_{corr} , as this naturally suggests to refer to the "correlated spin flips" there discussed.

25) Typos "correlate" -> "correlated" below (A1) and "an"->"a" above (A6).

Response to the Referee:

We appreciate these comments and have implemented them in the revised manuscript.

The Referee writes:

26) In the Equation (B16) which value of parameter "r" did the Authors use? Is it the 2.5 suggested below Eq. (B8)? Please clarify.

Response to the Referee:

We used, in fact, the factor 2.5. We are more explicit about this in the revised manuscript.

The Referee writes:

27) In the discussion of Appendix C2, and the case of transverse noise with no error-correction, the Authors should note that the numerical constant c (defined in the left column of page 17) depends also on the value of the estimated parameter ω [arxiv version of Ref [18] and doi:10.1103/PhysRevX.5.031010]. In particular, it vanishes at $\omega=0$ for which the exact Heisenberg scaling can be achieved [doi:10.1103/PhysRevX.5.031010].

Response to the Referee:

We thank the reviewer for this remark. We added a clarifying sentence in appendix C2 and included a reference to [J. B. Brask, R. Chaves, and J. Kołodyński, Phys. Rev. X 5, 031010 (2015).]

Response to Reviewer #2:

The Referee writes:

The manuscript is a thorough study into how bit or phase flip errors on a logical qubit can be corrected via a dissipative scheme. The process constitutes a dissipative version of the three qubit repetition code. The authors also discuss how such correction can be applied to quantum sensing.

The manuscript considers a self-correction process, but only one of limited scope. Only three qubits are used in the code, and only one type of error is corrected. This means that the best correction operator can be determined by looking at only two syndromes. It is such properties that allow their scheme to be implemented.

As such, I don't see that the work will be of great interest to those working of quantum error correction or self-correction for large scale devices. Instead it will interest those interested in error mitigation techniques in smaller scale set-ups, such as the quantum sensing discussed at the end of the manuscript.

I believe the manuscript is technically correct and of high quality. It is certainly suitable for publication in a good journal. However, I must defer to the opinion of someone within the quantum sensing community to determine whether it is of sufficient impact for Nature Communications.

Response to the Referee:

We thank referee #2 for reviewing our manuscript and for his/her assessment that our manuscript is "of high quality" and "suitable for publication in a good journal".

We present the first concrete experimental proposal for using engineered dissipation for improved quantum sensing. This work represents a first step into a new direction ultimately aiming for self-correction in future large-scale devices. Our focus lies on applications in quantum metrology (where only specific errors are targeted while complete error correction is undesirable). We nonetheless believe that it will inspire future work towards the goal of self-corrected quantum information processing, with a wide range of applications. We therefore think that our work will be of interest to a broad community.

As a first step into this new direction, we provide a guideline for developing robust small-scale quantum devices with current technology. The word "robust" is here not used in the sense of "fault-tolerant", since the main interest is the development of microscopic self-stabilization techniques that remove only specific types of errors that are a major concern for the considered applications.

We remark that it is in principle possible to correct for all Pauli errors by applying our scheme in a stroboscopic sequence that corrects x , y , and z errors in an alternating sequence x,y,z,x,y,z,\dots . We emphasize however, that this is not the aim of our paper since we focus on the dissipative improvement of quantum sensing (and as pointed out by the referee, correcting all Pauli errors simultaneously would be incompatible with this application, since this would result in a removal of the signal).

We are hopeful that the assessment of reviewer #1 who writes

"I find the presented results very interesting, with a clear potential to start a new direction in theoretical research of designing quantum error correction (QEC) protocols biased towards realistic experimental setups"

and reviewer #3 who states

"The proposed dissipative method for correcting spin-flip errors (and its extensions to correct phase-flip errors and correlated errors) is an important step towards reliable quantum sensing"

provide further evidence that the impact of our work warrants publication in Nature Communication. Indeed we believe that these reports provide an assessment from a quantum metrology expert.

Response to Reviewer #3:

The Referee writes:

The manuscript by Reiter et al. "Autonomous Quantum Error Correction and Application to Quantum Sensing with Trapped Ions" is very interesting, novel and exciting.

Trapped ions is a mature technology both for quantum computing/quantum information applications and for precision measurements, including the frequency standards. The proposed dissipative method for correcting bit-flip errors (and its extensions to correct phase-flip errors and correlated errors) is an important step towards reliable quantum sensing, as well as possible avenue towards scalable quantum computing.

Response to the Referee:

We are grateful to the Referee for this positive and encouraging evaluation of our work.

The Referee writes:

One drawback that I see in the proposed scheme as an experimentalist is the necessity to cool down to the ground state. While nearly-achievable, ground state cooling is never perfect, and heating is inevitable during operation time (especially if the interrogation time is long). Since vibrational modes play such an important role in the scheme, it would be informative to see the effects of imperfect cooling and subsequent heating mentioned in the paper. Would continuous sympathetic cooling be an useful option.

Overall, the paper is extremely well written and referenced, and I recommend its publication provided that my above question is addressed.

Response to the Referee:

We thank the Referee for this question, which has given us the chance to investigate further the experimental potential of our work:

To address the effects of imperfect cooling and motional heating we have performed additional numerical simulations. Here we have used a time-dependent master equation where we included the couplings to the auxiliary modes ($H_{\{aux,g\}}$ in the revised manuscript) and the cooling of the auxiliary modes (L_{κ}). In principle this cooling could also be modeled by introducing effective engineered cooling operators $L_{eng}^{(j)}$. For completeness, however, we chose to investigate the full problem including the full coupling to the auxiliary modes $c^{(j)}$.

In addition, we included heating operators, $L_{r^{\dagger}} = \sqrt{\kappa_r} r^{\dagger}$, for $r=a,b,c^{(1)}, c^{(2)}, c^{(3)}$. The Hilbert space of our simulation was extended to two excitations for each motional mode (including a third one did not change the results noticeably). For the motional modes, we assumed an initial thermal distribution (quasi-thermal, to be precise, given the truncation) of the population of the phononic states, described by a mean phonon number \bar{n} . For the other parameters we chose a paradigmatic case with a moderate ratio between the available sideband coupling (G) and the noise rate (Γ) with $G/\Gamma = 1000$. For the realistic assumption of a sideband coupling constant

$G/(2\pi) = 10$ kHz [personal communication with Philipp Schindler and Esteban Martinez] this corresponds to correction of individual spin-flips with a rate $\Gamma/(2\pi)=10$ Hz. In the absence of further imperfections, this yields a fidelity of $F=0.81$ for $t=1/\Gamma$, as compared to $F=0.56$ in the uncorrected single-ion case (see figure “ideal case” below). Note that the time-dependence of the simulation has allowed us to use bichromatic driving fields (i.e., two tones with opposite detunings $+G$ and $-G$ and identical Rabi frequencies), which reduces the error in this parameter regime. The driving strength (Ω), the sideband coupling to the auxiliary modes (g), and their decay rate (κ), as well as the available detunings were optimized numerically in the absence of external imperfections. Using this simulation, we investigated the effect of (A) imperfect cooling and (B) heating for (i) the auxiliary modes $c^{(j)}$ and (ii) the interrogation modes a and b . For (A) we assumed an initial $\bar{n}>0$ and no heating/cooling rates, and for (B) we included both cooling and heating operators, and the corresponding \bar{n} (determined by $\bar{n}=[\kappa^+]/[\kappa + \kappa^+]$). The results of our study are as follows:

For the auxiliary modes $c^{(j)}$ we find that the scheme is robust to both imperfect cooling and heating. This is due to the fact that the scheme already requires a strong cooling of the auxiliary modes to efficiently remove the errors from the system. In our simulation, we determine the optimum of the cooling rate to be about $\kappa \sim 0.86 \cdot G$ (the exact value is not critical). For $G/(2\pi) = 10$ kHz, we thus have $\kappa/(2\pi) = 8.6$ kHz, or a $1/e$ cooling time of the order of ~ 14 μ s. Such strong cooling rate for the auxiliary modes is technologically feasible using sympathetic cooling (see, e.g., [Y. Lin et al., Phys. Rev. Lett. 110, 153002 (2013)]). The cooling then overdamps the sideband coupling to the auxiliary modes (g , which has optimal value $\sim 0.2 G$), which justifies their adiabatic elimination. This resulting engineered decay process $L_{\text{eng}}^{(j)}$ used in the manuscript thus constitute a good approximation.

Using these parameters in our simulation, and assuming a heating rate of 10 quanta/second (similar to [P. Schindler et al., New J. Phys. 15, 123012 (2013)]) for each of the three auxiliary modes, as well as an additional cooling imperfection of $\bar{n}=0.01$ for each mode, led to a minor reduction of the fidelity of the codeword by 0.003 at $t = 1 / \Gamma$ (see figure “heating of the auxiliary modes” below). Given that a strong cooling rate is essential for the correction process also in the absence of heating, our results with respect to heating/imperfect cooling of the auxiliary modes do not impose any additional requirements on the system.

Imperfect cooling of the interrogation modes a and b is also found to only cause a moderate effect: Since the scheme is designed to operate in the ground-state manifold, errors are not corrected when the motion is in a higher excited state. Initial population in higher motional manifold should thus be seen as excluded from the error correction, constituting an additional error which is given by the motionally excited population. For small mean phonon number $\bar{n} \ll 1$, this population can be estimated by $P_{>0} \sim \bar{n}$ for each interrogation mode, a and b . Assuming the same mean phonon number for both modes, the resulting error is given by $P = P_{\{a\}} + P_{\{b\}} \approx 2 \cdot \bar{n}$. State of the art cooling techniques allow for small $\bar{n} \sim 0.01$. The resulting error due to imperfect cooling of the interrogation modes is thus much below the initial fidelity drop caused by the intrinsic error processes of the scheme, which, for the assumed parameters, is about $E_0 \sim 0.1$.

In contrast, the effect of heating of the interrogation modes is more pronounced, as these open an additional loss channel from the logical subspace (comparable to the effect of parallel noise discussed in the Imperfections section). This is due to the fact that the sideband coupling ($H_{\text{int},G}$ in the revised version) of the interrogation modes is not overdamped like the one of the auxiliary modes, but

is needed to create the splitting of the resonances. Motional excitations can thus lead to excitations of the ions, resulting in additional loss channels from the codespace. Similar effects have previously been analyzed in Lin et al., Nature 504, 415-418 (2013), Suppl. Inf. Sec. F.

The above effects can be counteracted by cooling the interrogation modes. For continuous cooling, the cooling rate should be moderate to minimize decoherence between the two paths of the scheme involving mode a and b . Alternatively, the cooling of the interrogation modes and error correction can be separated in time.

We address the imperfections of the interrogation modes numerically, using a motional cooling rate $\kappa = \Gamma$, and a heating rate of 3 quanta/second [C. Hempel, Doctoral Thesis, University of Innsbruck, 2014] for each of the modes a and b , and the corresponding thermal distribution with $\bar{n} = 0.05$. With these parameters, our simulation gives an additional error of 0.07 for $t = 1/\Gamma$, or a fidelity of $F \sim 0.74$, respectively (see figure “heating of the interrogation modes” below). As expected, the additional error due to heating of the interrogation modes is thus larger than that due to heating of the auxiliary modes, and is comparable to the initial drop of $E_0 \sim 0.1$. Yet, the operation of the error-correcting scheme still allows for much higher fidelity than in the uncorrected single-ion case with $F \sim 0.56$ at $t = 1/\Gamma$ and is thus useful for quantum metrology applications. The fidelity can be further increased by using stronger sideband couplings compared to the error rate.

We have included these results in our revised manuscript, adding a paragraph in the section “External imperfections” and a section to the Supplementary Information.

REVIEWERS' COMMENTS:

Reviewer #1 (Remarks to the Author):

I have carefully gone through the revised version of the manuscript and must admit that the Authors have thoroughly resolved all the issues I have raised in my previous report. While amending the text accordingly, they have also included extensive answers in their response to the points I have put forward (in particular, the possibility of performing parity measurements and the setting of non-perpendicular dephasing noise-type), for which I am very thankful.

Hence, I am left to do nothing but strongly recommend the work for publication in Nature Communications. I have spotted only minor mistakes and oversights which I list below. Nevertheless, I am convinced that the Authors can consider/resolve these themselves during the next steps of the editorial process—without the need of me inspecting the changes to the manuscript they further decide to make.

- 1) The Authors may want to be aware that during the period of reviewing the manuscript, the results of Ref. [24] about possibility of restoring Heisenberg Limit in metrology via (standard) error correction have been now generalised beyond the case of qubits in works: arXiv:1704.06280 and arXiv:1706.02445.
- 2) In Fig.4 the auxiliary modes of the first qubit should be labelled $c^{(1)}$ and not c_1 , to make now everything consistent with the text.
- 3) Let me emphasize that I very much appreciate the idea included now in the footnote [57] about the possibility for the proposed scheme to work even when the parameter encoding is directed along an arbitrary direction \vec{n} . However, let me just say (beyond the scope of the manuscript) that I have the intuition that applying such fast control pulses may, in principle, strongly (?) alter the interaction Lindblad operators (and their time-scales) assumed throughout the work, so I see that actually there "may be dragons" when directly supplementing the proposed scheme with dynamical-decoupling/control techniques. This idea would clearly require a separate project beyond the scope of the results presented.
- 4) Within the last part of the appendix, i.e., the Section IVB discussing the impact of parallel noise on the metrology scheme, the Authors should remind the reader about the definitions of local, Γ_z , and collective, Γ_Z , parallel dephasing they consider, and explicitly say which

case is actually plotted in Fig. S3. Also in my opinion the statement that the "prolongation of the Ramsey time is less effective for longer times $\Gamma\tau_R \geq 10^2$ " in the caption is a bit too optimistic... In fact, the plot shows that in such a regime it is actually better not to correct the errors! This should be briefly mentioned in the text of the appendix and, please, include also a comment that the "ideal" error correction makes in such a regime things even worse than the imperfect one—the reader should not be surprised that the data-point represented by the "square" in the plot is then actually below the one represented by the "triangle" at $\tau=10^2$.

Best regards.

Reviewer #3 (Remarks to the Author):

I would like to thank the authors for carefully investigating the effects of the experimental imperfections on the fidelity of their proposed scheme, as well as for additional analysis suggested by other referees. The manuscript is very solid and detailed, and the proposed scheme is novel and exciting, and actually implementable in a realistic setup. I strongly recommend its publication in Nature Communications.

Response to Reviewer #1:

The Referee writes:

I have carefully gone through the revised version of the manuscript and must admit that the Authors have thoroughly resolved all the issues I have raised in my previous report. While amending the text accordingly, they have also included extensive answers in their response to the points I have put forward (in particular, the possibility of performing parity measurements and the setting of non-perpendicular dephasing noise-type), for which I am very thankful.

Hence, I am left to do nothing but strongly recommend the work for publication in Nature Communications. I have spotted only minor mistakes and oversights which I list below. Nevertheless, I am convinced that the Authors can consider/resolve these themselves during the next steps of the editorial process—without the need of me inspecting the changes to the manuscript they further decide to make.

Response to the Referee:

We are grateful for this thorough and positive assessment of our revised manuscript, as well as for the additional comments, which we have implemented as detailed below.

The Referee writes:

1) The Authors may want to be aware that during the period of reviewing the manuscript, the results of Ref. [24] about possibility of restoring Heisenberg Limit in metrology via (standard) error correction have been now generalised beyond the case of qubits in works: arXiv:1704.06280 and arXiv:1706.02445.

Response to the Referee:

We thank the referee for these relevant references and have added them to our manuscript.

The Referee writes:

2) In Fig.4 the auxiliary modes of the first qubit should be labelled $c^{(1)}$ and not c_1 , to make now everything consistent with the text.

Response to the Referee:

We have adjusted the notation.

The Referee writes:

3) Let me emphasize that I very much appreciate the idea included now in the footnote [57] about the possibility for the proposed scheme to work even when the parameter encoding is directed along an arbitrary direction \vec{n} . However, let me just say (beyond the scope of the manuscript) that I have the intuition that applying such fast control pulses may, in principle, strongly (?) alter the interaction Lindblad operators (and their time-scales) assumed throughout the work, so I see that actually there "may be dragons" when directly supplementing the proposed scheme with dynamical-decoupling/control techniques. This idea would clearly require a separate project beyond the scope of the results presented.

Response to the Referee:

We have removed the footnote.

The Referee writes:

4) Within the last part of the appendix, i.e., the Section IVB discussing the impact of parallel noise on the metrology scheme, the Authors should remind the reader about the definitions of local, Γ_z , and collective, Γ_Z , parallel dephasing they consider, and explicitly say which case is actually plotted in Fig. S3. Also in my opinion the statement that the "prolongation of the Ramsey time is less effective for longer times $\Gamma\tau_R \geq 10^2$ " in the caption is a bit too optimistic... In fact, the plot shows that in such a regime it is actually better not to correct the errors! This should be briefly mentioned in the text of the appendix and, please, include also a comment that the "ideal" error correction makes in such a regime things even worse than the imperfect one—the reader should not be surprised that the data-point represented by the "square" in the plot is then actually below the one represented by the "triangle" at $\tau=10^2$.

Response to the Referee:

We have added a brief discussion of the types of errors relevant to that section and added the suggested statements.

Response to Reviewer #3:**The Referee writes:**

I would like to thank the authors for carefully investigating the effects of the experimental imperfections on the fidelity of their proposed scheme, as well as for additional analysis suggested by other referees. The manuscript is very solid and detailed, and the proposed scheme is novel and exciting, and actually implementable in a realistic setup. I strongly recommend its publication in Nature Communications.

Response to the Referee:

We are grateful to the referee for this positive judgement of our work and the encouraging feedback on the feasibility of our scheme.